# Live imaging and conditional disruption of native PCP activity using endogenously tagged zebrafish sfGFP-Vangl2

Maria Jussila [1], Curtis W. Boswell [1,2,3], Nigel W. Griffiths[1], Patrick G. Pumputis [1,2] & Brian Ciruna [1,2] ✉

Tissue-wide coordination of polarized cytoskeletal organization and cell behaviour, critical for normal development, is controlled by asymmetric membrane localization of non-canonical Wnt/planar cell polarity (PCP) signalling components. Understanding the dynamic regulation of PCP thus requires visualization of these polarity proteins in vivo. Here we utilize CRISPR/Cas9 genome editing to introduce a fluorescent reporter onto the core PCP component, Vangl2, in zebrafish. Through live imaging of endogenous sfGFP-Vangl2 expression, we report on the authentic regulation of vertebrate PCP during embryogenesis. Furthermore, we couple sfGFP-Vangl2 with conditional zGrad GFP-nanobody degradation methodologies to interrogate tissue-specific functions for PCP. Remarkably, loss of Vangl2 in *foxj1a*-positive cell lineages causes ependymal cell cilia and Reissner fiber formation defects as well as idiopathic-like scoliosis. Together, our studies provide crucial insights into the establishment and maintenance of vertebrate PCP and create a powerful experimental paradigm for investigating post-embryonic and tissue-specific functions for Vangl2 in development and disease.

The non-canonical Wnt/planar cell polarity (PCP) signalling pathway controls polarized, uniform orientation of cells within a plane of a tissue. The PCP pathway regulates a number of developmental and homeostatic processes by directing collective asymmetric modification of the cell cytoskeleton, leading groups of cells to either move or divide in a shared direction, or position their organelles such as the basal body asymmetrically[1–5]. Molecular genetic studies in *Drosophila* have identified essential roles for asymmetric membrane localization of opposing PCP signalling proteins in both the establishment and intercellular propagation of polarity[6]. Despite identification of a multitude of PCP-regulated morphogenetic processes, characterization of vertebrate PCP at the cellular and molecular level is still lacking. This is challenging, because detailed observation requires live imaging capabilities and the ability to differentiate asymmetric protein localization across membranes of adjacent cells.

Historically, immunohistochemical studies have provided snapshots of PCP within tissues[7,8] while exogenous, fluorescently-tagged reporter proteins have been introduced for mosaic imaging of PCP. Although great complexities in the dynamic regulation of vertebrate PCP have been reported across tissues and polarized cell behaviours[9–12], the functionality and spatiotemporal accuracy of these static and exogenous readouts remain uncertain. The biological relevance of studies utilizing exogenous, fluorescently-tagged PCP molecules also remains to be determined, as overexpression of PCP proteins can disrupt cell polarity[11,13] and functionality of published PCP fusion proteins has not been validated via rescue of orthologous mutant phenotypes[12,14–16]. Furthermore, non-physiological expression of exogenous proteins can mask subtle localization patterns or signalling dynamics. This is highlighted by observations that in many occasions, co-expression of another PCP component is required for

[1]Program in Developmental & Stem Cell Biology, The Hospital for Sick Children, 686 Bay Street, Toronto, ON M5G 0A4, Canada. [2]Department of Molecular Genetics, The University of Toronto, Toronto, ON M5S 1A8, Canada. [3]Present address: Department of Genetics, Yale University School of Medicine, New Haven, CT 06510, USA. ✉e-mail: ciruna@sickkids.ca

asymmetric membrane localization of the fluorescent fusion protein of interest[17–20].

Recent advances in genome editing technologies have facilitated the generation of fluorescently-tagged knock-in alleles and, to date, endogenously tagged PCP molecules have been reported in both *C. elegans* and mouse[21,22]. However, methods to live-image PCP during mouse embryonic development are limited, especially at the level of mosaically-labelled single cell clones. Therefore, to understand the dynamic regulation of vertebrate PCP, we turned to zebrafish where the optical transparency of externally developing embryos enables powerful embryonic cell transplantation and live-imaging methodologies. Specifically, we aimed to image and manipulate the endogenous zebrafish Vangl2 protein, a core PCP component and essential regulator of vertebrate planar polarity[23,24]. Notably, tagging mouse Vangl2 with a tdTomato fluorophore resulted in hypomorphic protein activity that caused variable neural tube closure defects[22]. Therefore, using CRISPR/Cas9 gene editing, we introduced a superfolder GFP (sfGFP, a bright and rapidly maturing fluorophore) onto the N-terminus of zebrafish Vangl2 to generate a functional and native PCP reporter protein.

In this work we perform detailed embryonic cell transplantation and single-cell level analyses to document the dynamic sub-cellular localization of sfGFP-Vangl2 over the course of neural tube formation, revealing intriguing intercellular associations of Vangl2 positive cell membranes with polarized basal bodies in floorplate cells. Furthermore, we exploit zGrad[25] GFP-specific protein degradation methodologies to conditionally interrogate sfGFP-Vangl2 function in floor plate cells, demonstrating an essential role for Vangl2 in both the establishment and maintenance of polarized basal body positioning as well as an inherent and surprising directionality to the intercellular propagation of PCP. Finally, we show that degradation of sfGFP-Vangl2 in motile-ciliated cell lineages causes ependymal cell cilia and Reissner fiber formation defects, as well as idiopathic-like scoliosis. Our work highlights the importance and utility of studying endogenous proteins in the context of PCP and opens the door to post-embryonic and tissue-specific analysis of zebrafish Vangl2 function in organogenesis, tissue homeostasis as well as the pathogenesis of congenital malformations and disease states.

## Results and discussion

### A functional membrane-localized sfGFP-Vangl2 fusion protein

To visualize the dynamic regulation of endogenous PCP signalling in zebrafish, we used CRISPR/Cas9 genome editing to target a superfolder green fluorescent protein (sfGFP) onto the N-terminus of Vangl2, an essential and membrane-localized PCP signalling protein (Fig. 1a; targeting details in Methods section). Embryos carrying the knock-in allele showed prominent sfGFP signal in the brain and spinal cord at 28 h post-fertilization (hpf) (Fig. 1b). In striking contrast to the *vangl2*[m209/209] loss-of-function mutant phenotypes, which include a shortened body axis due to abnormal PCP-driven convergence and extension movements as well as embryonic lethality[23], fish homozygous for the *vangl2*[sfGFP] allele or trans-heterozygote for *vangl2*[sfGFP] and *vangl2*[m209] alleles are viable, fertile and appear morphologically normal (Fig. 1c). Closer inspection revealed that homozygote *vangl2*[sfGFP/sfGFP] embryos are slightly shorter than wildtype siblings at 24 hpf (Fig. 1d). Interestingly, heterozygote *vangl2*[m209/+] embryos are similarly shorter than wildtype, which has not been previously reported. *Vangl2*[sfGFP/m209] transheterozygote embryos demonstrate a further shortening of the body axis but are significantly less affected than the *vangl2*[m209/m209] mutants (Fig. 1d). Altogether these data indicate that, although mildly hypomorphic, the sfGFP-Vangl2 fusion protein is functional and can therefore be used to interrogate endogenous PCP signalling dynamics.

To determine the onset of Vangl2 expression, we imaged *vangl2*[sfGFP/sfGFP] embryos (hereafter referred to as *vangl2*[sfGFP]) through blastula and gastrula stages. Weak Vangl2 membrane localization

could be observed in a subset of cells as early as high stage (3.3 hpf), which became more ubiquitous through early epiboly stages (4.6 hpf; Fig. 1e). Interestingly, weak Vangl2 membrane-localization was also observed within enveloping layer (EVL) cells at these stages (Supplementary Fig. 1a). These observations are consistent with a role for Vangl2 prior to gastrulation, perhaps in promoting tissue cohesion in margin cells during early blastoderm morphogenesis[26]. Vangl2 expression levels increased, and membrane localization became more prominent with the onset of gastrulation at shield stage, persisting through 75% epiboly and bud stages (Fig. 1e, Supplementary Fig. 1b–e). This corresponds to the known role for Vangl2 in the regulation of convergence and extension movements[23]. When compared to immunohistochemical analyses of Vangl2 expression[14], the superior consistency, sensitivity and live-imaging capabilities of sfGFP-Vangl2 expression highlight its utility for experimental studies.

### Endosomal trafficking of Vangl2

In addition to membrane-localized Vangl2, we also observed punctate intracellular accumulations, which became prominent at the onset of gastrulation as large cytosolic aggregates (Fig. 1e). To investigate the identity of these compartments, we analysed sfGFP-Vangl2 localization in relation to known markers of intracellular sorting organelles (Fig. 1f–i and Supplementary Figs. 1b–e, 2a–c). Prior to gastrulation the zebrafish Golgi is fragmented[27], and whereas we observed occasional sfGFP-Vangl2 signal in the proximity of the trans-Golgi network reporter GalT-RFP, Vangl2 was predominantly localized elsewhere in the cell (Fig. 1i and Supplementary Figs. 1e, 2b). While we observed that Vangl2 puncta colocalized to some extent with all endosomal markers (Fig. 1f–h and Supplementary Fig. 1b–d), quantification revealed that a majority of Vangl2 puncta were either Rab5c or Rab7 positive, while fewer were Rab11a positive (Supplementary Fig. 2b), suggesting that Vangl2 is mainly trafficked through endosomal transport to lysosomes and to a lesser extent recycled back to membranes. Interestingly, the largest Vangl2 puncta were particularly enriched for Rab7 colocalization (Supplementary Fig. 2c).

As introduction of a fluorescent tag may affect protein structure and a recently published mouse tdTomato-Vangl2 knock-in allele was shown to cause neural tube closure and other developmental defects[22], we also considered that observed localization of sfGFP-Vangl2 to late endosomes might represent autophagosome targeting of misfolded protein. However, immunofluorescent labeling of endogenous LC3-positive autophagosomes revealed colocalization with only a fraction of Vangl2 puncta (Supplementary Fig. 2a–c). Because cytoplasmic sfGFP-Vangl2 puncta appear only transiently in development, disappearing at later embryonic stages (Fig. 2a; see below), this suggests that they do not represent an innate misfolded protein state, but may rather reflect a normal developmental process. In accordance, immunohistochemical analysis of endogenous Vangl2 localization has also identified cytoplasmic punctate staining at gastrula stages[14]. Overall, we cannot exclude the possibility that the sfGFP tag induces some level of misfolding and degradation of Vangl2 during gastrulation. However, because homozygous knock-in zebrafish are viable and appear grossly normal, we conclude that the sfGFP-tag does not significantly compromise Vangl2 protein folding or activity.

### Dynamic anterior neuroepithelial cell enrichment of Vangl2

Following gastrulation, strong Vangl2 expression was observed in the central nervous system, whereas only weak sfGFP-Vangl2 signal could be detected in surrounding axial and paraxial mesodermal lineages (Figs. 1b, 2a). Previous work has demonstrated that exogenous GFP-tagged Vangl2 or its cytoplasmic binding partner Prickle polarize anteriorly in neuroepithelial cells in zebrafish and *Xenopus*[9,12,14,16,28,29]. To characterize the asymmetric distribution of endogenous Vangl2, we performed cell transplantation at blastoderm stages to generate clones of *vangl2*[sfGFP] cells within wildtype host embryos and imaged

Vangl2 localization in individual neuroepithelial cells at multiple time points during neural tube formation (Fig. 2b). At all stages, weak sfGFP-Vangl2 signal could be observed across the entire neuroepithelial cell membrane, with bright membrane sfGFP-Vangl2 domains enriched in puncta at anterior cell surfaces (Fig. 2c, Supplementary Fig. 3a and Supplementary Movie 1). Vangl2 localization varied considerably

between cells, including apical and basal distribution patterns, as well as localization into cell protrusions (Fig. 2c and Supplementary Fig. 3a). Previously, exogenous Vangl2 reporter localization has been quantified along single cellular planes. To gain a global view of Vangl2 distribution, we performed 3-dimensional quantification of sfGFP-Vangl2 membrane-enrichment in comparison to an RFP membrane reporter

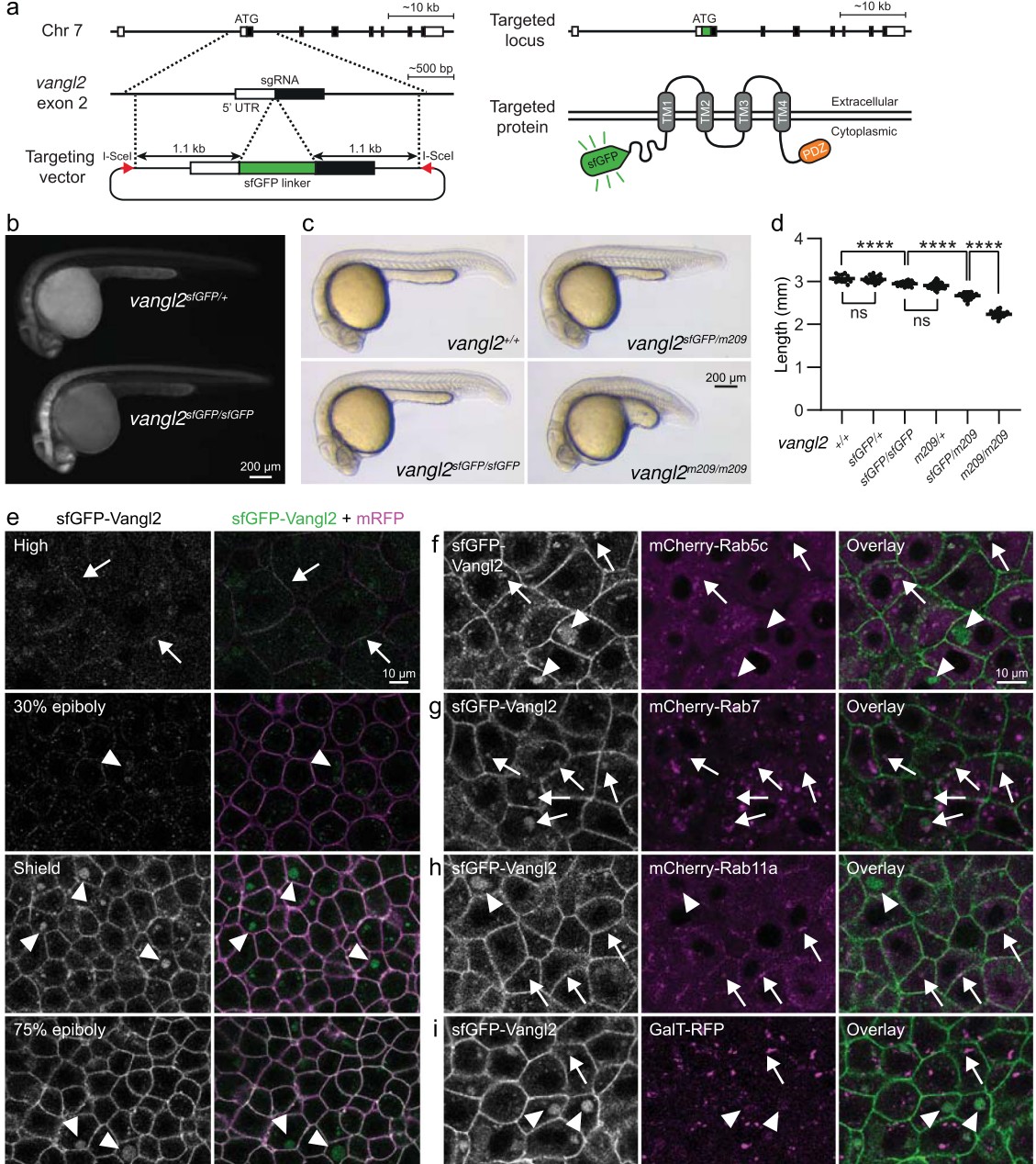

**Fig. 1 | sfGFP-Vangl2 is functional and dynamically localized to membranes during early zebrafish embryogenesis. a** A schematic illustration of the sfGFP-Vangl2 knock-in targeting strategy. **b** Wide-field fluorescence images demonstrating sfGFP-Vangl2 expression in *vangl2^{sfGFP/+}* and *vangl2^{sfGFP/sfGFP}* embryos at 28 h post-fertilization (hpf). **c** Lateral view of wild-type, *vangl2^{sfGFP/sfGFP}*, *vangl2^{sfGFP/m209}* and *vangl2^{m209/m209}* embryos at 24 hpf. **d** Embryo length quantification at 24 hpf. Data shows representative experiments combined from independent crosses that were repeated 2-4 times. Statistical analysis was performed using one-way ANOVA with Tukey's multiple comparisons test (****$P < 0.0001$). Wild-type $n = 14$, *vangl2^{sfGFP/+}* $n = 22$, *vangl2^{sfGFP/sfGFP}* $n = 20$, *vangl2^{m209/+}* $n = 82$, *vangl2^{sfGFP/m209}* $n = 72$, *vangl2^{m209}* $n = 63$. Horizontal line labels mean, and dots indicate individual embryo measurements. Source data are provided as a Source Data file. **e** Live confocal images of

representative *vangl2^{sfGFP/sfGFP}* (hereafter referred as *vangl2^{sfGFP}*) embryos at blastula through early gastrula stages, as indicated ($n = 6$ for each stage). Ectodermal cells were imaged in gastrulating embryos. Embryos were injected with mRNA coding for a membrane-localized monomeric RFP (mRFP) reporter. All sfGFP images were acquired using identical settings. Arrows point at membrane-localized Vangl2 at high stage and arrowheads point at cytoplasmic Vangl2 puncta. Live confocal images of representative shield staged *vangl2^{sfGFP}* embryos injected with mRNA coding for mCherry-Rab5c (**f**), mCherry-Rab7 (**g**), mCherry-Rab11a (**h**) or GalT-RFP (**i**) reporter constructs ($n = 4$ for each reporter). Arrows point at cytoplasmic Vangl2 puncta in close proximity to respective reporters, and arrowheads point at isolated Vangl2 puncta.

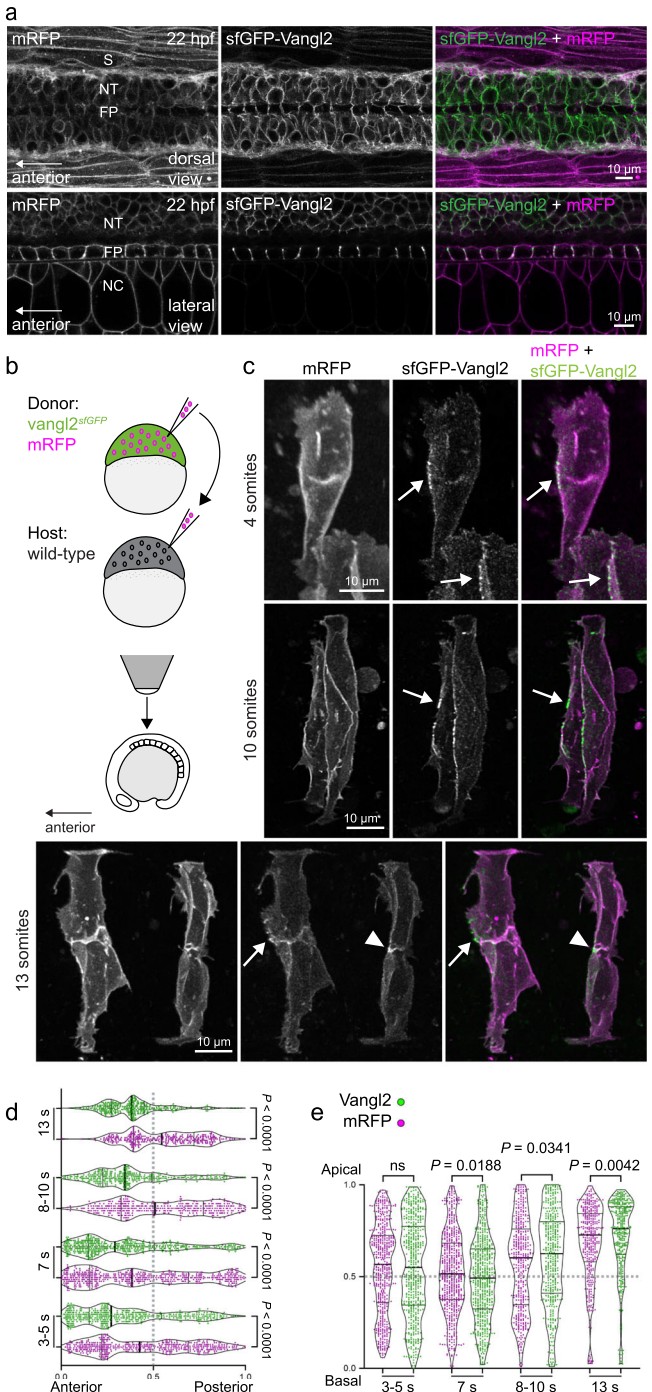

**Fig. 2 | Vangl2 is enriched in a planar polarized manner on anterior neuroepithelial membranes. a** Live confocal images comparing sfGFP-Vangl2 and membrane-RFP localization at 22 h post fertilization (hpf) in dorsal (top) and lateral (bottom) orientations. Anterior is to the left in all images. Strong Vangl2 expression is observed on cell membranes in the neural tube (NT) including the floor plate (FP), while only weak expression is observed in somites (S) and notochord (NC). A representative image from $n = 3$ embryos shown. **b** A schematic illustrating transplantation of membrane-RFP-labelled *vangl2sfGPP* donor cells into WT (wild-type) host embryos, at sphere stage. Chimeric embryos were imaged dorsally, at positions between the 2nd and 8th somite. **c** Representative confocal images of membrane-RFP and sfGFP-Vangl2 localization in the developing spinal cord of chimeric embryos at 4-somite, 10-somite and 13-somite stages of development, as quantified in (**d**, **e**). Maximum intensity projections are shown. Arrows point to Vangl2 enrichment on anterior neuroepithelial cell membranes, and arrowheads to anterior apical membranes at the neural midline. Anterior is to the left in all images. Distribution of the brightest sfGFP-Vangl2 and membrane-RFP spots along anterior-posterior (**d**) and apical-basal (**e**) axes of *vangl2sfGFP* neuroepithelial cells, quantified at four consecutive stages of neural tube morphogenesis. Data from multiple transplanted *vangl2sfGFP* cells was pooled (3-5 somites, $n = 7$; 7 somites, $n = 7$; 8-10 somites, $n = 6$; 13 somites, $n = 5$) and axial lengths normalized to 1. Details of the quantification can be found in the Methods section. Statistical analysis was performed using a two-tailed Mann-Whitney test. Thicker line shows median, and thinner lines first and third quartile. Source data are provided as a Source Data file.

studies (predominantly using exogenous reporters) that PCP components concentrate into potential "signalosomes" on the cell membrane[9,16,20,22,30–34]. In *Xenopus*, exogenous Vangl2 and Prickle reporters accumulate with actomyosin at shrinking neuroepithelial cell junctions[9] and, in mesodermal cells undergoing convergence and extension movements, Prickle2 colocalizes with the LifeAct filamentous actin reporter[34,35]. These observations suggest that Vangl2/PCP directly influences cytoskeletal re-organization. To investigate the relationship between anterior Vangl2 enrichment and actin organization, we co-expressed a mCherry-LifeAct reporter in *vangl2sfGFP* cells. Live imaging revealed that the majority of neuroepithelial cell protrusive activity and LifeAct localization was restricted to apical and basal surfaces, and not to anterior membranes where sfGFP-Vangl2 was enriched (Supplementary Movies 2, 3). However, sfGFP-Vangl2 could be observed accumulating within active cellular protrusions at the anterior membrane (Supplementary Fig. 3a, Supplementary Movie 4), consistent with known roles for PCP and Vangl2 in regulating filopodia and cell protrusive activity during gastrulation and facial branchiomotor neuron migration[11,12,36]. Together, although the presence of bright sfGFP-Vangl2 puncta supports the notion that PCP signalling may function in specific subdomains on the anterior membrane, the presence of Vangl2 membrane enrichment was not predictive of a particular membrane behaviour.

## Vangl2 polarity is regulated non-cell autonomously

To further understand how neighbouring cells affect asymmetric Vangl2 localization into different membrane-domains, we performed cell transplantation experiments to generate clones of *vangl2sfGFP* cells within a *vangl2tkSOf* loss-of-function host embryo (Fig. 3a). In this PCP-deficient environment, *vangl2sfGFP* cells clustered together, suggesting that Vangl2 expressing cells may recognize and preferentially interact with one other. While Vangl2 remained localized to the cell membrane, planar polarized membrane enrichment was lost and Vangl2 appeared to concentrate at cell protrusions (Fig. 3d and Supplementary Fig. 3b). We picked a subset of the most mediolaterally aligned cells to quantify asymmetric Vangl2 localization and found no difference in the distribution of the brightest Vangl2 and mRFP spots along the anterior-posterior or apico-basal axes (Fig. 3b, c; Supplementary Fig. 4c, d). Loss of polarity in *vangl2sfGFP* positive clones supports a cell non-autonomous regulation of vertebrate PCP. The persistence of Vangl2 in cell protrusions suggests that it might function in mediating local

(mRFP) using the spot function in Imaris (detailed description in Methods). We plotted the subset of membrane spots containing the highest sfGFP-Vangl2 and mRFP fluorescent intensities across anterior-posterior and apico-basal cell axes (Supplementary Fig. 4a, b) and found that, despite the variation between cells, Vangl2 was significantly enriched at the anterior membrane at all stages analysed (Fig. 2d). Vangl2 localization was less uniformly polarized along the apico-basal axis, with Vangl2 showing strongest apical enrichment at the 13 somite-stage concomitant to neural tube midline formation[16] (Fig. 2e).

To examine the dynamics of Vangl2 protein localization, we performed time-lapse imaging of *vangl2sfGFP* chimeric embryos between 10-13 somite stages. Persistent sfGFP-Vangl2 enrichment was observed along anterior membrane domains (Supplementary Movie 1), supporting evidence from multiple *Drosophila*, *Xenopus* and mouse

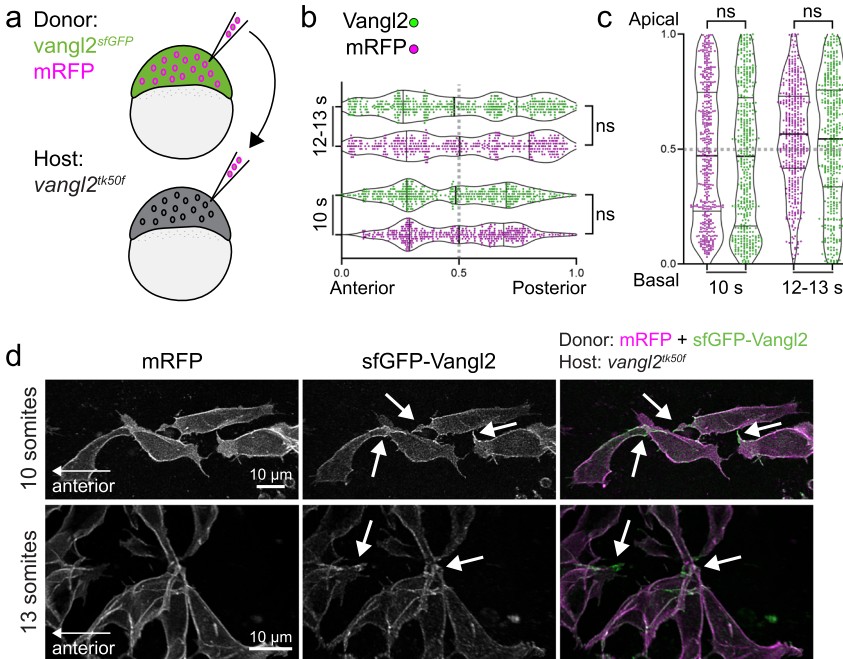

**Fig. 3 | Planar polarized enrichment of Vangl2 is controlled by cell non-autonomous PCP signalling. a** A schematic illustrating transplantation of membrane-RFP-labelled *vangl2^sfGFP* donor cells into *vangl2^tk50f* loss-of-function host embryos, at sphere stage. **b, c** Distribution of the brightest sfGFP-Vangl2 and membrane-RFP spots along anterior-posterior (**g**) and apical-basal (**h**) axes of *vangl2^sfGFP* neuroepithelial cells transplanted into a *vangl2^tk50f* mutant host, quantified at two consecutive stages of neural tube morphogenesis. Data from multiple transplanted *vangl2^sfGFP* cells was pooled (10 somites, *n* = 7; 12-13 somites, *n* = 9).

Statistical analysis was performed using a two-tailed Mann-Whitney test. Thicker line shows median, and thinner lines first and third quartile. Source data are provided as a Source Data file. **d** Representative confocal images of membrane-RFP and sfGFP-Vangl2 localization in *vangl2^sfGFP* neuroepithelial cells within *vangl2^tk50f* mutant hosts, at 10-somite and 13-somite stages of development, as quantified in **b, c**. Maximum intensity projections are shown. Arrows point at Vangl2 enrichment on cell protrusions. Anterior is to the left in all images.

signalling that is independent of its role in global anterior-posterior neural tube planar polarity.

The re-organization and re-establishment of cell polarity over the course of cell division remains a poorly understood feature of vertebrate PCP. In zebrafish, asymmetric localization of exogenous Prickle-GFP reporter molecules is lost upon neuroepithelial cell division, but is re-established as daughter cells reintegrate into the neuroepithelium[16]. In mouse epidermis, different studies have shown PCP components to be internalized during mitosis or to persist on cell membranes in a polarized manner[15,37]. Exogenous Vangl2, specifically, has been shown to persist on cell membranes and to get trans-internalized from neighbouring cell membranes into the dividing cell[10]. To determine the fate of endogenous Vangl2 during mitosis, we imaged *vangl2^sfGFP* neuroepithelial cells over the course of cell division. Notably, asymmetric sfGFP-Vangl2 distribution was lost upon cell rounding, but Vangl2 remained at the cell membrane (Supplementary Movies 3, 4). We observed sfGFP-Vangl2 enriched at the poles of dividing cells, as well as to basolateral membrane projections that are maintained during division (Supplementary Movies 3, 4). Although some intracellular sfGFP-Vangl2 signal was observed in dividing cells, we did not observe significant endocytosis of membrane Vangl2 or trans-endocytosis between neighbouring cells (Supplementary Movies 3, 4). Reacquisition of Vangl2 polarization in daughter cells was not obvious immediately upon division (Supplementary Movie 2). Of interest, sfGFP-Vangl2 positive membrane protrusions from distant neighbours could be observed interacting with dividing cells (Supplementary Movie 4). Out of 24 cells that had an anterior Vangl2 positive protrusion, 11 were touching a dividing cell, and the same was observed for 15/32 apical protrusions (total *n* = 112 cells). Our results show that Vangl2 remains on the membrane but loses its asymmetry during mitosis, and that re-establishment of cell polarity may be regulated by contacts from neighbouring cells.

## Exogenous Wnt signals do not perturb neuroepithelial PCP

The identity of the upstream polarizing cue for PCP remains a long-standing question in the field. Although Wnt ligands are not required for establishment of PCP in *Drosophila*[38,39], vertebrate studies suggest that non-canonical Wnt ligands are both required for normal PCP activity and are able to instruct the directionality of planar polarity[16,18,40–43]. To determine whether exogenous sources of Wnt5b and/or Wnt11, two Wnt ligands essential for zebrafish PCP[40,41], could instruct PCP of neighboring cells in the neuroepithelium, we utilized a combination of embryonic cell transplantation and transient transgenic approaches. Briefly, *vangl2^sfGFP* cells were transplanted into host embryos that harbored mosaic integration of a heatshock-inducible Wnt expression transgene (Supplementary Fig. 5a, b). These transgenes included a bicistronic IRES-tdTomato reporter cassette to monitor and identify cellular sources of Wnt5b or Wnt11 expression. Experimental embryos were heat-shocked at the onset of neural tube morphogenesis, and effects on sfGFP-Vangl2 polarity were analyzed 5-6 h later (Supplementary Fig. 5b–d). We quantified the posterior:anterior ratio sfGFP-Vangl2 membrane localization in *vangl2^sfGFP* cells that were situated anterior or posterior to neighbouring tdTomato-positive Wnt5b or Wnt11 expressing cells but observed no difference in sfGFP-Vangl2 polarity when compared to controls (Supplementary Fig. 5e). Although it was not possible to measure the strength of exogenous Wnt5b or Wnt11 expression, nor their effects on local Wnt gradients, our observations do not support an instructive role for Wnt ligands on zebrafish neuroepithelial PCP at stages assayed.

## Polarized intercellular association of Vangl2 and cilia

In addition to cytoskeletal organization, polarized basal body (BB) positioning has been identified as a conserved readout of PCP from insects to vertebrates[44]. In zebrafish floorplate cells, BBs are localized to the posterior apical membrane by 24 hpf, and the establishment of

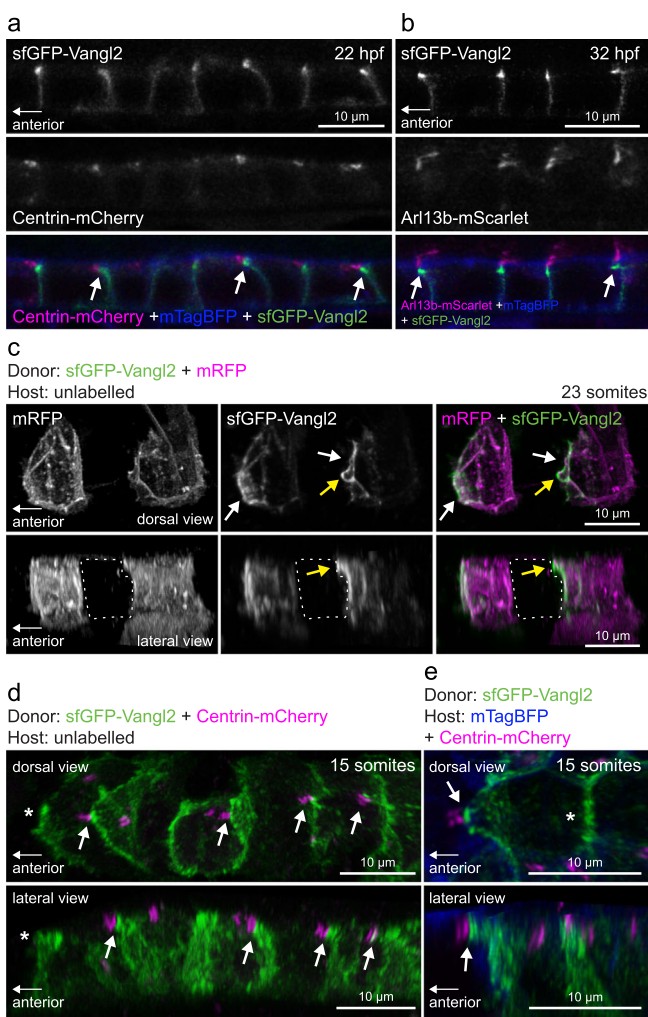

**Fig. 4 | Vangl2-positive anterior membrane closely associates with planar polarized basal bodies of floor plate cells. a** Live confocal images of sfGFP-Vangl2 localization in relation to basal body (Centrin-mCherry) and membrane (mTagBFP-CAAX) reporters within floorplate cells of a *vangl2^sfGPP* embryo at 22hpf. Lateral view, anterior to the left. Arrows point at sfGFP-Vangl2 enriched membranes extending apically towards anterior neighbouring cells. A representative image from *n* = 3 embryos shown. **b** Live confocal images of sfGFP-Vangl2 localization in relation to cilia axoneme (Arl13b-mScarlet) and membrane (mTagBFP-CAAX) reporters within floorplate cells of a *vangl2^sfGPP* embryo at 32hpf. Lateral view, anterior is to the left. Arrows point at sfGFP-Vangl2 enriched membranes extending apically towards anterior neighbouring cells. A representative image from *n* = 5 embryos shown. **c** Live confocal image comparing sfGFP-Vangl2 and membrane-RFP localization in floorplate cells of a 23-somite staged chimeric embryo (*mRFP* mRNA-injected *vangl2^sfGFP* cells transplanted into WT hosts). Maximum intensity projections, dorsal (top) and lateral (bottom) views are shown, anterior is to the left. White arrows point at Vangl2 enrichment on anterior cell membranes. Yellow arrows point at Vangl2 enriched membrane protruding from the anterior apical surface. Dashed line highlights an unlabelled host floorplate cell between two donor cells. A representative image from *n* = 16 cells shown. **d** Live confocal image comparing sfGFP-Vangl2 and Centrin-mCherry localization in floorplate cells of a 15-somite staged chimeric embryo (*Centrin-mCherry* mRNA-injected *vangl2^sfGFP* cells transplanted into WT hosts). Maximum intensity projections, dorsal (top) and lateral (bottom) views are shown, anterior to the left. Arrows point at Vangl2 on the anterior membrane that is closely associated with a basal body docked on the posterior membrane. Asterisk indicates anterior unlabelled host cell that is not Centrin-mCherry positive. A representative image from *n* = 18 cells shown. **e** Live confocal image comparing sfGFP-Vangl2 localization in relation to basal body (Centrin-mCherry) and membrane (mTagBFP) reporters in neighbouring floorplate cells of a 15-somite staged chimeric embryo (*vangl2^sfGFP* cells transplanted into WT hosts injected with *Centrin-mCherry* and *mTagBFP-CAAX* mRNA). Maximum intensity projections, dorsal (top) and lateral (bottom) views are shown, anterior to the left. Arrow points at Vangl2 on the anterior membrane of a donor cell that is closely associated with a basal body docked on the posterior membrane of a host cell. Asterisk indicates lack of Centrin-mCherry labelled basal body in the donor cell. A representative image from *n* = 17 cells shown.

BB polarity requires Vangl2[13,29,45]. To examine the relationship between Vangl2 localization and BB positioning, we imaged the floorplate of *vangl2^sfGFP* embryos at 22 and 32 hpf. In the sagittal plane, sfGFP-Vangl2 was enriched on the anterior apical cell membrane (Fig. 4a, b), in accordance with observations using an exogenous GFP-Vangl2 transgenic reporter[12,29]. However, in some floorplate cells apically enriched Vangl2 appeared to extend anteriorly towards neighbouring cells (Fig. 4a, b). This bright apical Vangl2 signal was closely associated with BBs labelled with Centrin-mCherry, and with Arl13b-positive cilia (Fig. 4a, b). We imaged individual transplanted *vangl2^sfGFP* floorplate cells and found that Vangl2 was indeed localized asymmetrically on anterior cell membranes, and that in some cells it further localized to an anterior apical projection (Fig. 4c). Labelling BBs in transplanted *vangl2^sfGFP* cells with Centrin-mCherry confirmed the proximity of anterior Vangl2 with posteriorly polarized BBs (Fig. 4d). Furthermore, labelling of host cell BBs confirmed that Vangl2-positive apical membrane closely associated with the posterior BB of neighbouring cells (Fig. 4e, 11/17 cells). Of interest, exogenous Frizzled3a, a putative intercellular binding partner of Vangl2, has been shown to be enriched posteriorly into puncta at the base of cilia in zebrafish[29]. We posit that Vangl2-mediated intercellular interactions could therefore exert non-cell autonomous effects on cytoskeletal organization, apical polarity and/or cell shape changes to influence the polarized site of BB docking or posterior tilting of the floorplate motile cilia[13,45,46].

### Conditional loss of sfGFP-Vangl2 reveals role in BB polarity

To determine whether Vangl2 also controls the maintenance of BB polarity, we took advantage of zGrad GFP-nanobody protein

degradation methodologies to conditionally manipulate endogenous sfGFP-Vangl2 protein levels[25]. We first confirmed that zGrad is non-toxic to embryos, and that *zGrad* mRNA-injected *vangl2^sfGFP/sfGFP* embryos phenocopy *vangl2* loss-of-function mutants (Supplementary Fig. 6a, b). Furthermore, because *vangl2* mutants are subject to maternal effects[16], we investigated the consequences of zGrad mRNA-injection across a full genotypic spectrum of maternal and zygotic *vangl2^sfGFP* alleles. Strikingly, sequential zGrad-mediated degradation of maternal and zygotic *vangl2^sfGFP* gene products yielded progressively more severe phenotypes as measured by embryo length and axial length-width ratios (Fig. 5a, Supplementary Fig. 6c), recapitulating maternal and zygotic *vangl2* mutant phenotypes. Analysis of protein levels confirmed a complete elimination of sfGFP-Vangl2 by zGrad at 24 hpf (Fig. 5b). This highlights the sensitivity and functionality of sfGFP-Vangl2 zGrad protein degradation methods.

To conditionally manipulate sfGFP-Vangl2 protein levels within the floorplate, we generated a ubiquitously expressed *Tg(β-actin2::loxP-mCherry-STOP-loxP-zGrad)* transgene (Fig. 5c) and crossed it to the *Tg(foxj1a::iCre)* line[47], which expresses Cre recombinase (activating zGrad) specifically within motile-ciliated cell lineages. At 32 hpf, when BB polarity has been established, sfGFP-Vangl2 was still present at the membrane of experimental *vangl2^sfGFP/sfGFP;Tg(βactin2::loxP-mCherry-STOP-loxP-zGrad);Tg(foxj1a::iCre)* floorplate cells. However, mosaic loss of sfGFP-Vangl2 could be observed by 48 hpf (Fig. 5d), allowing us to analyse both cell-autonomous and non-autonomous roles for Vangl2 in the maintenance of BB polarity. Images were scored for the loss of Vangl2 in individual cells and their immediate anterior and posterior neighbours. Irrespective of their Vangl2 expression status, cells surrounded by sfGFP-negative neighbours demonstrated a loss of BB polarity when compared to control populations. Strikingly, the presence of sfGFP-Vangl2 in a posterior neighbouring cell was associated

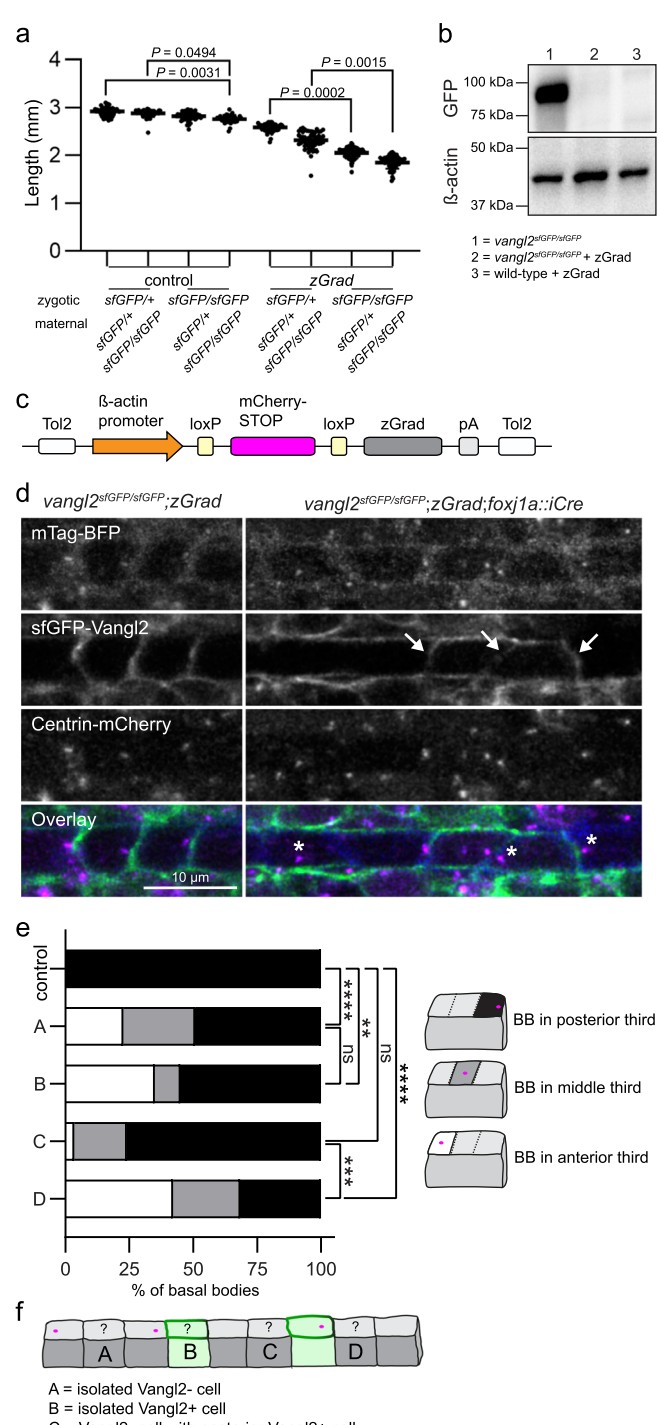

**Fig. 5 | Conditional degradation of sfGFP-Vangl2 reveals a cell non-autonomous requirement for Vangl2 in maintaining basal body polarity. a** Quantification of embryo lengths for *zGrad* mRNA-injected and control embryos at 24 hpf. Maternal and zygotic genotypes are as indicated. Data is pooled from three independent experiments. Control n from left to right: *n* = 53, *n* = 62, *n* = 61, *n* = 36. zGrad *n* from left to right: *n* = 55, *n* = 51, *n* = 53, *n* = 54. Horizontal line labels mean, and dots indicate individual embryo measurements. Statistical analysis was performed using Kruskal-Wallis test with Dunn's multiple comparisons test. Source data are provided as a Source Data file. **b** A representative Western blot detecting sfGFP-Vangl2 protein levels using a GFP antibody in uninjected *vangl2^sfGFP/sfGFP* embryos and *vangl2^sfGFP/sfGFP* or wild-type embryos injected with 50 pg of *zGrad* mRNA. β-acting is used as a loading control. *N* = 3; 50 uninjected *vangl2^sfGFP/sfGFP*, 50 *zGrad* injected wild-type, and 70 *zGrad* injected *vangl2^sfGFP/sfGFP* embryos were lysed per sample. **c** A schematic of the conditional zGrad transgene. **d** Live confocal images of sfGFP-Vangl2 expression in relation to basal body (Centrin-mCherry) and membrane (mTagBFP) reporters within floorplate cells of control *vangl2^sfGFP/sfGFP*;Tg(βactin2::loxP-mCherry-STOP-loxP-zGrad)* embryos (left) and experimental *vangl2^sfGFP/sfGFP*;Tg(βactin2::loxP-mCherry-STOP-loxP-zGrad);Tg(foxj1a::iCre)* embryos (right) at 48 hpf. Dorsal view, anterior to the left. Arrows indicate residual sfGFP-Vangl2 expression in a subset of floorplate cells following *zGrad* expression. Asterisks indicate mispolarized basal bodies following *zGrad* expression. **e** Quantification of basal body localization within the posterior, medial or anterior thirds of floor plate cells for control and *vangl2^sfGFP/sfGFP*;Tg(βactin2::loxP-mCherry-STOP-loxP-zGrad);Tg(foxj1a::cre)* embryos described in **b**. Cells were categorized as A-D based on the presence or absence of sfGFP-Vangl2 signal within both the cell and its immediate neighbours, as illustrated in **f**. Control, *n* = 164; A, *n* = 256; B, *n* = 20; C, *n* = 29; D, *n* = 19. Statistical analysis was performed using Kruskal-Wallis test with Dunn's multiple comparisons test. Source data are provided as a Source Data file. **f** A schematic illustrating cell categories quantified in **e**. Grey colour indicates cells that have lost sfGFP-Vangl2 signal; green colour indicates residual sfGFP-Vangl2 expression.

dpf (Fig. 6a–e). The curvatures varied in severity, perhaps reflecting mosaic nature in the timing of Vangl2 loss. Embryos exhibiting moderate to severe curves failed to inflate their swim bladders by 5 dpf and did not survive beyond embryonic stages (Fig. 6b, c; 62%; *n* = 139/223 embryos). Remarkably, embryos exhibiting mild curvatures, as well as a subset of moderate curves (Fig. 6c–e; 38%, *n* = 84/223 embryos), survived to adulthood and 100% of these fish developed obvious spinal curvatures (Fig. 6f–h). Spinal curvatures appeared independent of embryonic defects, as conditional mutant embryos with no visible embryonic phenotype (Fig. 6e) also developed scoliosis at juvenile stages (Fig. 6i; *n* = 7/9). Micro-CT imaging of 2-month-old conditional mutant fish revealed rotational spinal deformities that occurred in the absence of congenital vertebral malformations (Fig. 6f–i; *n* = 25), closely resembling multiple zebrafish models of adolescent idiopathic scoliosis[48–52]. Control *vangl2^sfGFP/sfGFP*;Tg(βactin2::loxP-mCherry-STOP-loxP-zGrad)* animals (which lacked Cre recombinase expression) did not develop scoliosis (Fig. 6f; *n* = 134), nor did *vangl2^sfGFP/+*;Tg(βactin2::loxP-mCherry-STOP-loxP-zGrad);Tg(foxj1a::iCre)* fish that experienced only heterozygous loss of sfGFP-Vangl2 (*n* = 85). Additionally, we did not detect scoliosis in fish heterozygous for the *vangl2^m209* loss-of-function allele, nor in *vangl2^sfGFP/m209* trans-heterozygote fish. These observations demonstrate that the scoliosis phenotype results from degradation of sfGFP-Vangl2 protein in the absence of wild-type Vangl2, specifically within FoxJ1a-positive motile ciliated cell lineages.

**Mutants show ependymal cell cilia and Reissner fiber defects**
Notably, conditional loss of *ptk7a*, a critical regulator of Wnt signal transduction, from *foxj1a*-positive lineages within the brain has also been demonstrated to cause idiopathic scoliosis (IS)[47]. However, because Ptk7a regulates both non-canonical Wnt/PCP and canonical Wnt/β-catenin signalling, the molecular basis underlying scoliosis phenotypes remained unknown. To determine whether Ptk7a and Vangl2 regulate spine morphogenesis via a common mechanism, we examined *vangl2^sfGFP/sfGFP*;Tg(βactin2::loxP-mCherry-STOP-loxP-zGrad);Tg(foxj1a::iCre)* fish for two phenotypes that have been linked to

with restored BB polarity in Vangl2-negative cells, even if the posterior neighbour demonstrated abnormal BB positioning (Fig. 5e, f). In contrast, the presence of sfGFP-Vangl2 in anterior neighbours did not restore polarity of Vangl2 negative cells. These results demonstrate that Vangl2 is required for the maintenance of BB polarity, and that Vangl2 functions non-cell autonomously to propagate PCP in a posterior to anterior direction.

**Loss of Vangl2 in motile-ciliated lineages causes scoliosis**
In addition to the basal body phenotype, the majority of *vangl2^sfGFP/sfGFP*;Tg(βactin2::loxP-mCherry-STOP-loxP-zGrad);Tg(foxj1a::iCre)* embryos developed variable axial curvatures mainly in the dorsal direction by 3

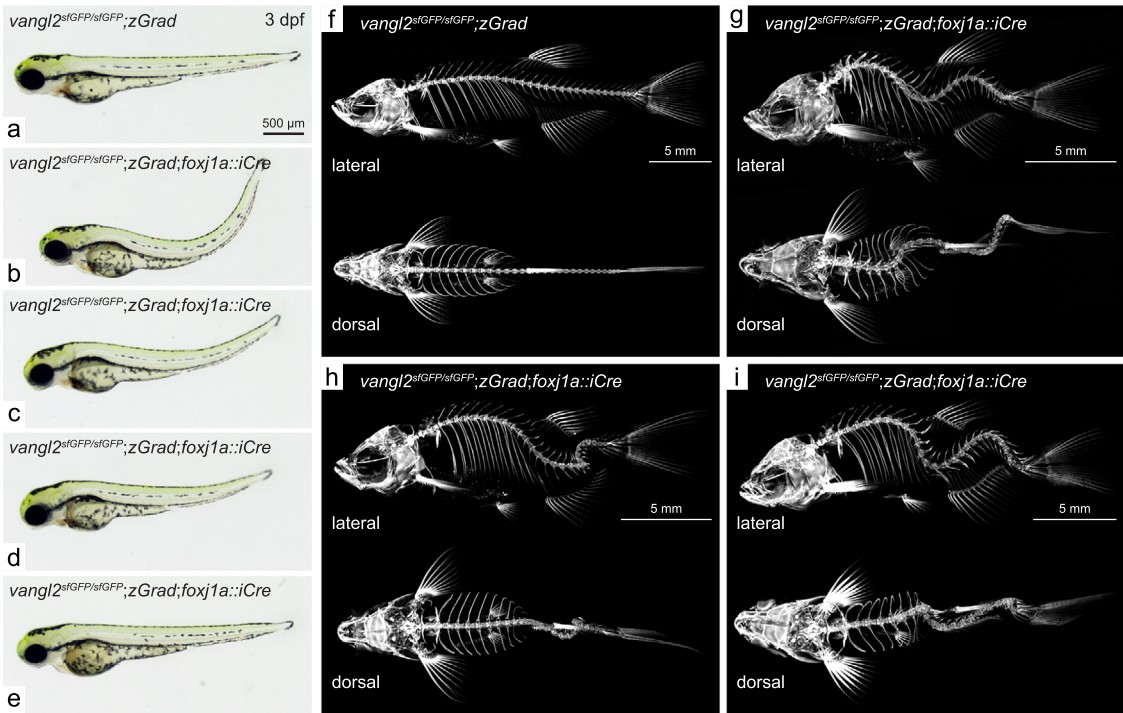

**Fig. 6 | Degradation of sfGFP-Vangl2 in *foxj1a*-positive cell lineages causes embryonic axial curvature and postembryonic idiopathic-like scoliosis.** Overview of body axis phenotypes observed in control (**a**) and *vangl2^sfGFP/sfGFP*;Tg(βactin2::loxP-mCherry-STOP-loxP-zGrad);Tg(foxj1a::iCre)* embryos (**b**–**e**) at 3 dpf. Lateral and dorsal projections of 3-dimensional micro-CT scans of 2 month old control (**f**) and *vangl2^sfGFP/sfGFP*;Tg(βactin2::loxP-mCherry-STOP-loxP-zGrad);Tg(foxj1a::iCre)* fish (**g**–**i**).

scoliosis in *ptk7a* mutant models: brain ependymal cell cilia defects[49] and loss of Reissner fiber formation[50]. Defects in brain ependymal cell (EC) cilia function are believed to modulate cerebral spinal fluid flow and have been associated with idiopathic-like scoliosis in *ptk7a* mutants and other zebrafish IS models[47,49,52]. To determine whether conditional *vangl2^sfGFP/sfGFP* mutants also display defects in EC cilia formation, we performed scanning electron microscopy (SEM) of the rhombencephalic ventricle of brains dissected from adult *vangl2^sfGFP/sfGFP*;Tg(βactin2::loxP-mCherry-STOP-loxP-zGrad);Tg(foxj1a::iCre)* experimental animals as well as *vangl2^sfGFP/sfGFP*;Tg(βactin2::loxP-mCherry-STOP-loxP-zGrad)* sibling controls. While control animals exhibited a dense mat of EC cilia lining the brain ventricle (Fig. 7a, b; *n* = 8), conditional *vangl2^sfGFP/sfGFP* mutant brains exhibited sparse or absent EC cilia phenotypes (Fig. 7c, d; *n* = 8) that closely resembled those described for *ptk7a* mutant zebrafish[49]. Reissner fiber (RF) is a proteinaceous filament that originates from the subcomissural organ, threads through the ventricles of the brain and spinal cord, and is thought to control CSF homeostasis[53]. Loss of Scospondin (Sspo), the main protein component of RF, is sufficient to cause IS in zebrafish and the loss of RF and/or ectopic Sspo aggregation has been linked to scoliosis in zebrafish *ptk7a* IS models[50,51]. To determine whether conditional *vangl2^sfGFP/sfGFP* mutants also display RF formation defects, we analyzed Sspo localization in control and mutant brains at 21 dpf when spinal curves have begun to develop (Fig. 8a, b). Similar to *ptk7a* fish, *vangl2^sfGFP/sfGFP*;Tg(βactin2::loxP-mCherry-STOP-loxP-zGrad);Tg(foxj1a::iCre)* mutant brains exhibited abnormal accumulation of Sspo in forebrain (Fig. 8c, d) and rhombencephalic (Fig. 8e, f) ventricles, as well as an increased immunoreactivity in the area where RF normally forms. Taken together, these results suggest that scoliosis in conditional *vangl2^sfGFP/sfGFP* mutants may be mechanistically connected to IS in *ptk7a* mutant model. This indicates a key role for PCP signalling in spine morphogenesis and supports growing evidence that mutations in core PCP genes *PTK7* and *VANGL1*, identified in human patients with adolescent idiopathic scoliosis, may be functionally linked with spinal curvature[48,54,55].

In summary, our work provides essential insights into the establishment and maintenance of vertebrate PCP. Live imaging of endogenous Vangl2 has authenticated broad principles predicted using exogenously introduced reporter molecules, including organization of PCP across the embryonic anterior-posterior axis and concentration of core PCP proteins into 'signalosomes' at the cell membrane. It has also yielded intriguing findings that (1) demonstrate dynamic redistribution of Vangl2 within discrete subcellular membrane domains over the course of gastrulation and neurulation, (2) suggest tissue-specific contexts for published tenets regarding membrane recycling of PCP proteins during mitosis or their association with the actin cytoskeleton, (3) indicate an inherent and surprising directionality to the intercellular propagation of PCP, and (4) establish a critical role for Vangl2 within motile-ciliated cell lineages for normal zebrafish spine development, possibly linking PCP defects with the pathogenesis of idiopathic scoliosis. Outside of mouse and *C. elegans*[21,22], functional fluorescently-tagged PCP knock-in alleles have not been widely reported. Our *vangl2^sfGFP* allele thus represents an invaluable resource that, when combined with embryonic cell-transplantation, live-imaging and conditional zGrad degradation methodologies, establishes a powerful experimental paradigm for future investigations into vertebrate cell polarity.

## Methods

### Animal husbandry

Zebrafish husbandry and experimental protocols were approved by the Hospital for Sick Children's Animal Care Committee, and all protocols were performed in accordance with Canadian Council on Animal Care guidelines. Wild-type zebrafish from TU strains were used. The *vangl2* mutant allele *tri^tk50f* was used for cell transplantation experiments[23], and the *tri^m209* allele was used in to validate the *vangl2^sfGFP* allele functionality because of the ability to use a published

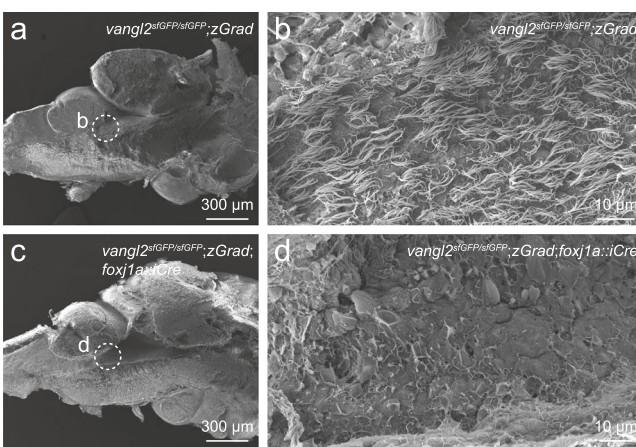

**Fig. 7 | FoxJ1a-lineage specific degradation of sfGFP-Vangl2 results in ependymal cell cilia defects.** Scanning electron micrographs of the rhombencephalic ventricle in brains dissected from 7-8 month old *vangl2^sfGFP/sfGFP*;*Tg(βactin2::loxP-mCherry-STOP-loxP-zGrad)* control (**a**, **b**; *n* = 8) and *vangl2^sfGFP/sfGFP*;*Tg(βactin2::loxP-mCherry-STOP-loxP-zGrad);Tg(foxj1a::iCre)* mutant (**c**, **d**; *n* = 8) zebrafish. Dashed circles in **a**, **c** indicate the area of high magnification images in **b**, **d**.

PCR-mediated genotyping strategy[23,42]. *Tg(foxj1a::iCre)* line has been previously described[47]. Embryos from natural matings were grown at 28 °C unless otherwise indicated.

### sfGFP-Vangl2 targeting strategy

**Target selection.** A previously validated guide RNA (gRNA) was used to target the sequence surrounding the start codon of *vangl2* (targeting sequence GGATAACGAGTCGCAGTACTCGG)[56]. As a proof of principle, a solution containing 300 mM KCl, 100 ng *vangl2* gRNA, 200 ng recombinant Cas9 (PNA Bio cat. CP01) and 10% phenol red was incubated at 37 °C for 5 min to form ribonucleoprotein complexes (RNPs), and injected into wild-type embryos at the 1-cell stage. Embryos were (1) scored for *vangl2/trilobite* phenotypes at 28 hpf and (2) assessed for loss of a ScaI restriction site by PCR to gauge mutagenesis efficiency.

**Targeting vector construction.** Prior to vector construction, male and female TU wild-type animals were screened by PCR for polymorphisms surrounding *vangl2* exon 2. Left and right arms of homology of ~1-1.2 kb in length were amplified from fin clips and subjected for Sanger sequencing. Fish containing highly similar amplicons were selected for use in targeting experiments (4 males and 4 females). A ~2.2 kb amplicon surrounding *vangl2* exon 2 was amplified from these wild-type fish and cloned into pDONR221. A superfolder GFP (sfGFP) was amplified from sfGFP-N1 (a kind gift from Michael Davidson and Geoffrey Waldo, Addgene #54737)[57] and cloned into the *vangl2* targeting intermediate with a 3x(GGGGS) flexible linker by megaprimer PCR[58] and verified for in-frame inclusion. Site-directed mutagenesis was performed on this intermediate to abolish the *vangl2* gRNA targeting site to prevent vector from being cleaved in vivo. This final targeting intermediate was amplified and cloned into XhoI and EcoRI sites of pKHR5 (a kind gift from D. Grunwald, Addgene #74593). All targeting intermediates and final clones were fully verified by Sanger sequencing.

**Knock-in injections.** A 2.5 µl mix containing 20 ng of *vangl2* targeting vector, 1.25 U of I-SceI (NEB cat. R0694S), 0.5x Cutsmart buffer and 10% phenol red was prepared. A second 2.5 µl mix containing 300 mM KCl, 100 ng *vangl2* gRNA and 200 ng recombinant Cas9 (PNA Bio cat. CP01) was incubated at 37 °C for 5 min. The two mixes were combined and one nanolitre was injected into incrosses between pre-designated TU

wild-type animals. Mosaic GFP fluorescence was observed in several F0 embryos which were separated and grown to adulthood. Adult F0 animals were outcrossed to wildtype Tu fish and screened for GFP + fluorescence. A founder transmitting 36% of GFP + embryos was identified, and F1 embryos were screened by PCR using the following primers, F: CCGCGCTCTCCAGTCCGTCA, R: CGAGAGCTGCGTGAG TGTGAA, to ensure precise editing of the *vangl2* locus.

**Line establishment and maintenance.** The F1 fish were incrossed to produce homozygous F2 fish. Embryos were screened either visually by GFP fluorescence or genotyped using the primers listed above.

### Transgenesis

Transgenesis was performed using standard Gateway-compatible procedures[59]. The zGrad open reading frame was amplified from pCS2(+)-zGrad (a kind gift from H. Knaut; Addgene #119716) and cloned into pDONR-P2rP3 to generate p3E-zGrad via Gateway technology (Invitrogen). To generate *Tg(βactin2::loxP-mCherry-STOP-loxP-zGrad)* zebrafish, p5E-*βactin2*[58], pME-*loxP mCherry STOP loxP* (a gift from K. Kwan), and p3E-*zGrad* were recombined into pDEST Tol2 pA2 transgenesis vector. The *wnt5b* and *wnt11* open reading frames were amplified from embryonic cDNA using the following primers: Wnt5b F ATGGATGTGAGAATGAACC, Wnt5b R CTACTTGCACACAAACTGG, Wnt11 F ATGACAGAATACAGGAACTTTC and Wnt11 R TCACTTGCA-GACGTATCTCTCG. Both amplicons were gel extracted and recombined into pDONR221 via Gateway technology (Invitrogen) to generate pME-*wnt5b* and pME-*wnt11*. p3E-IRES-tdTomato was generated by replacing the eGFP ORF in p3E-*IRES-eGFP*[58] with tdTomato amplified from ptdTomato (Clonetech #632531). Finally, pME-*wnt5b* or pME-*wnt11* were recombined into pDEST Tol2 pA2 transgenesis vector with p3E-*IRES-tdTomato* and p5E-*hsp70*[59] to generate *Tg(hsp70::wnt5b-IRES-tdTomato)* and *Tg(hsp::70wnt11-IRES-tdTomato)*. Clones were fully verified by Sanger sequencing. Embryos were injected at the one-cell stage with 25 pg of assembled transgene and 25 pg of *Tol2* mRNA. Embryos were sorted at 48 hpf for mCherry reporter expression (zGrad) or γ-crystallin::GFP (Wnt5b and Wnt11) and were subsequently grown to adulthood. Individuals were bred to TU wild-type zebrafish to generate stable F1 lines. All stable transgenic lines used were hemizygous for the respective transgenes.

### Plasmids and embryo microinjections

Rab5c, 7 and 11a plasmids were a kind gift from B. Link, mCherry-LifeAct-7 was a kind gift from M. Davidson (Addgene #54491), pCS2(+)-zGrad was a gift from H. Knaut (Addgene #119716)[25], pME-*mTagBFP-CAAX* was a gift from N. Cole (Addgene #75149)[60], zebrafish codon-optimized mScarlet was a kind gift from T. Thiele, centrin-mCherry was a kind gift from A. Salic and GalT-RFP was a kind gift from J. Wallingford. N-terminal mCherry-linker was cloned into the Rab plasmids and mCherry-LifeAct and mTagBFP-CAAX were cloned into pCS2+ using Gateway technology (Thermo Fisher Scientific). Plasmids were linearized and sense-strand-capped mRNA was synthesized with the mMESSAGE mMACHINE system (Ambion). 15 pg membrane-localized monomeric RFP (mRFP), 20 pg mTagBFP-CAAX, 30 pg mCherry-LifeAct, 5-25 pg mCherry-Centrin, 2.5 pg Arl13b-mScarlet, 10 pg mCherry-Rab5c, 10 pg mCherry-Rab7, 15 pg mCherry-Rab11a, 10 pg GalT-RFP and 50-100 pg zGrad were injected at one-cell stage.

### Cell transplantations

Cell transplantations were performed at mid-blastula stages, as described previously[61,62]. Briefly, mRNA encoding fluorescent markers used in each experiment were injected into the donor or host embryos at one-cell stage. Embryos were dechorionated chemically using pronase (Sigma #P5417) prior to transplantations as the donors reached high-oblong and hosts oblong-sphere stage. Transplants were performed in embryo media using a glass capillary pipette and a

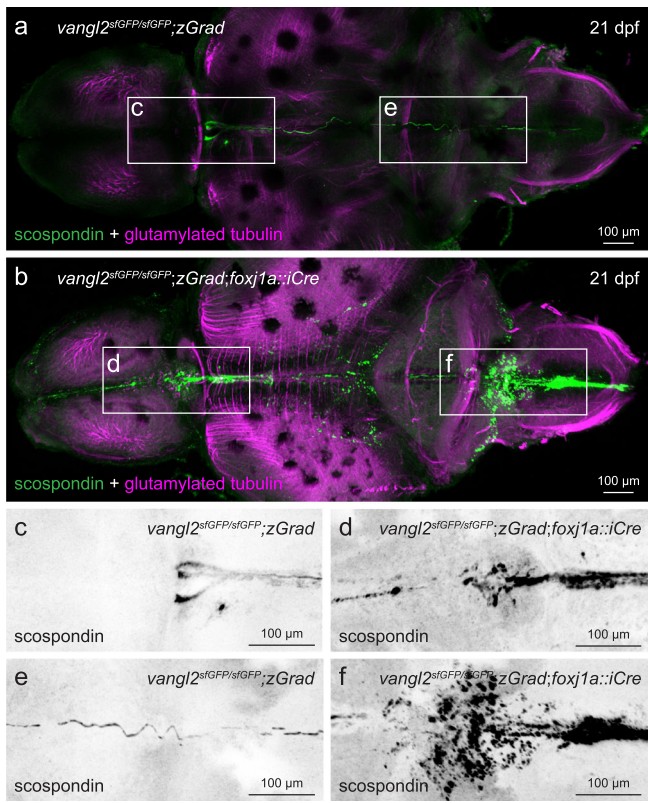

**Fig. 8 | FoxJ1a-lineage specific degradation of sfGFP-Vangl2 results in ectopic scospondin accumulation and disrupted Reissner fiber formation.**
**a**, **b** Representative maximum intensity Z-stack projections of confocal micrographs, acquired through dorsally-oriented whole mount brains that were dissected from 21dpf *vangl2^sfGFP/sfGFP*;*Tg(βactin2::loxP-mCherry-STOP-loxP-zGrad)* control (**a**; *n* = 4) and *vangl2^sfGFP/sfGFP*;*Tg(βactin2::loxP-mCherry-STOP-loxP-zGrad)*;*Tg(foxj1a::iCre)* mutant (**b**; *n* = 9) brains, and immunostained for polyglutamylated tubulin (magenta) and scospondin (green). Inverted, higher magnification images of scospondin immunostaining in forebrain (**c**, **d**) and rhombencephalic (**e**, **f**) ventricles of control (**c**, **e**) and mutant (**d**, **f**) brains, represented by boxed areas in **a**, **b**.

transplantation rig with an oil-filled Hamilton syringe. Cells some distance away from the embryonic margin were aspirated into the glass capillary and transplanted into hosts. For the early somite stage imaging, embryos were grown at 25 °C after transplantations.

## Immunofluorescence
Embryos were fixed at shield stage with 100% methanol for LC3 immunofluorescence. Samples were washed three times for 15 min in 1X PBS and 30 min in 1X PBS with 0.1% Tween-20 (PBST) at room temperature. Samples were blocked in 5% normal goat serum (NGS), 0.5% 1 mg/mL BSA in 1X PBST overnight for 2 h at room temperature. Primary antibodies were incubated on samples overnight at 4 °C in blocking solution, samples were washed with 1X PBST, and incubated with secondary antibodies overnight at 4 °C in blocking solution. Samples were washed with 1X PBST and mounted in 0.8 % low melt agarose for imaging. The following antibodies were used: anti-LC3, rabbit polyclonal (1:100, Novus Biologicals #NB100-2331SS, lot AM), Goat anti-Rabbit IgG (H + L) Alexa Fluor Plus 594 (1:1000, Thermo-Fisher Scientific, #A32740, lot UG288488).

21 dpf fish were euthanized and fixed in cold 100 % methanol for Reissner substance and glutamylated tubulin immunofluorescence. Samples were washed three times for 15 min in 1X PBS and 1 h and 30 min in 1X PBS with 0.1% Tween- 20 at room temperature. Samples were blocked in 5% normal goat serum (NGS), 0.5% 1 mg/mL BSA in 1X PBS with 0.1% Tween-20 overnight at 4 °C. Primary antibodies were

incubated for four nights at 4 °C in block (5% normal goat serum (NGS), 0.5% 1 mg/mL BSA in 1X PBS with 0.1% Tween-20). Secondary antibodies were incubated for three nights at 4 °C in block. Samples were washed with 1X PBST over three nights between and after antibody incubations. The following antibodies were used for immunohistochemistry: anti-Polyglutamylation Modification, mouse monoclonal (GT335) (1:500, Adipogen Lifesciences #AG-20B-0020-C100, lot A27791601), anti-Reissner substance, rabbit polyclonal[50] (1:500, gift from Stephane Gobron), Goat anti-Rabbit IgG (H + L) Alexa Fluor Plus 488 (1:1000, ThermoFisher Scientific #A32731, lot UI295697) and Goat anti-Mouse IgG (H + L) Alexa Fluor Plus 647 (1:1000, ThermoFisher Scientific; #A32728, lot WK331591). Brains were dissected at the end of the staining and cleared in glycerol.

## Western blot
Zebrafish embryos at were manually dechorionated at 24 hpf and deyolked by triturating embryos with a 200 μL pipette in cold Ringer's solution with 1 mM EDTA and 0.3 mM PMSF[63]. Deyolked embryos were then washed 3 times in cold Ringer's solution. RIPA lysis buffer with Halt protease and phosphatase inhibitor cocktail (ThermoFisher Scientific #78446) was used to lyse the embryos. Samples were homogenized with a microfuge pestle. Protein concentration in the lysate was measured using the Pierce BCA protein assay kit (ThermoFisher Scientific #23225). Equal amounts of total protein samples were resolved on a 10% Mini-Protean TGX gel (Bio-Rad #4561036) and transferred to a PVDF membrane (Bio-Rad #1620177). Membranes were then incubated with primary antibodies, rabbit anti-GFP (1:2000, ThermoFisher Scientific #A11122,lot 51527 A) and rabbit anti-β-actin (1:500, Cell Signaling Technology #4967, lot 12), overnight at 4°C on a light shake, followed by washes, then incubated for 1 h at room temperature with secondary antibodies, goat anti-rabbit IgG H&L (HRP) (1:15000, Abcam #ab6721, lot GR3357864-9). Finally, detection was performed on membranes with Clarity Western ECL Substrate (Bio-Rad #170-5060) and imaged on a ChemiDoc MP (Bio-Rad).

## Imaging
Brightfield embryonic imaging was performed on a Leica M80 microscope with MC170 HD camera (Leica). Wide-field fluorescence imaging was performed on a Zeiss AXIO Zoom V16 microscope (EMS3/SyCoP3, Zeiss) and ORCA-Flash 4.0 C11440-22C camera (Hamamatsu). Embryonic confocal imaging was performed on a Leica TCS SP8 Lightning confocal with Leica DMI8 microscope, 40x/1.3 water immersion objective and HyD detectors using the 405, 488 and 552 laser lines and Leica LAS X software (Leica). Samples were mounted in 0.8% low-melt agarose on glass-bottom dishes with a coverslip. The images were deconvolved using the Lightning module (Leica LAS X Lightning, Leica). All time-lapse imaging was performed using a resonant scanner with 16 times averaging. Maximum intensity projections and time lapse images were recorded using Imaris release 9.9.1 (Bitplane). Brain confocal imaging was performed on a Leica TCS SP8 Lightning confocal with Leica DMI8 microscope, 10x/0.4 air immersion objective and HyD detectors using the 552 and 638 laser lines and Leica LAS X software (Leica). Samples were cleared in glycerol and mounted on glass slides with coverslips. Images were prepared using Imaris Viewer version 9.9.1 (Bitplane), Photoshop release 23.4.2 (Adobe) and Illustrator release 26.4.1 (Adobe).

## Measurements of embryonic lengths and L/W ratio
Embryo lengths were measured by manually drawing a line from the front of the head to the tip of the tail in FIJI version 1.53f51 (ImageJ). Embryonic length/width ratios were calculated by measuring the widest and longest points of *krox20* and *myoD* expression. Statistical analysis was performed using one-way ANOVA with Tukey's multiple comparisons test in GraphPad Prism version 9.3.1 (GraphPad Software).

## Quantification of Vangl2 colocalization with cytoplasmic markers

Confocal images of live embryos injected with mCherry-Rab5c, mCherry-Rab7, mCherry-Rab11a or GalT-RFP, or fixed embryos immunostained for LC3, were segmented in Imaris release 9.9.1 (Bit-Plane). Three embryos were analysed for each marker, and three areas of 69.9 μm × 69.9 μm containing 3 consecutive slices (2.48 μm in depth) were analysed within each embryo. Signals in the green and red channels were segmented using the surfaces function. The following parameters were used for each marker (Smoothing/Background Sub-traction/Seed Points Diameter (Intensity Based): sfGFP-Vangl2 0.3/2/0.4, mCherry-Rab5c 0.3/0.6/0.7, mCherry-Rab7 0.3/2/0.6, mCherry-Rab11a 0.3/0.7/0.4, LC3 0.3/0.15/0.4. For GalT-RFP, 0.3 smoothing, absolute intensity with manual thresholding and 0.25 seed points diameter were used. Machine learning combined with manual selection was used to exclude the sfGFP membrane signal from cytoplasmic puncta. Cytoplasmic sfGFP-Vangl2 puncta that were touching were unified into one object and puncta smaller than 0.2 $\mu m^3$ were excluded from the analysis. Vangl2 puncta touching each respective marker were manually selected and colocalization was presented as their percentage of all Vangl2 puncta, or Vangl2 puncta bigger than 1.5 $\mu m^3$. Statistical analysis was performed using one-way ANOVA with Tukey's multiple comparisons test or two-tailed $t$-test in GraphPad Prism version 9.3.1 (GraphPad Software).

## Quantification of asymmetric Vangl2 membrane-localization

Comparison of Vangl2 and mRFP membrane enrichment was performed using the spot function in Imaris release 9.9.1 (BitPlane). In short, membrane-signals for mRFP or sfGFP were decorated with spots, and the spots for the two channels were merged. A reference point was placed in the centre of each cell to determine the anterior-posterior (X) and apical-basal (Y) axis. The position of each spot was determined as their distance to the reference point. Mean fluorescence intensity for mRFP and sfGFP within each spot was calculated, and the 10% of brightest spots for each fluorophore were plotted against their normalized X and Y positions using GraphPad Prism version 9.3.1 (GraphPad Software). Statistical analysis was performed using a two-tailed Mann-Whitney test in GraphPad Prism version 9.3.1 (GraphPad Software).

## Generation of mosaic Wnt expressing cells using heatshocks

Wild-type host embryos were injected with 20 pg *mTagBFP-CAAX* mRNA, 25 pg of *Tg(hsp70::wnt5b-IRES-tdTomato)* or *Tg(hsp70::wnt11-IRES-tdTomato)* transgene plasmid DNA and 25 pg of *Tol2* mRNA. Control embryos were injected with 20 pg *mTagBFP-CAAX* mRNA and 25 pg of *Tol2* mRNA. Cells from *vangl2^sfGFP* embryos were transplanted into transgene-injected hosts as described above. Transplanted embryos were heatshocked starting from 1-6 somite stages for 10 min at 37 °C for three times, with 30 min between each heatshock. Embryos were imaged 5-6 h after the last heatshock.

## Quantification of posterior/anterior sfGFP-Vangl2 ratio upon local Wnt expression

SfGFP-Vangl2 cells with a tdTomato-expressing cell (marking heatshock-induced *wnt5b* or *wnt11* expression) 3 cell-diameters away in either anterior or posterior direction, or variable Wnt5b or Wnt11 cells in multiple locations in close proximity, were selected for analysis. Any sfGFP-Vangl2 cells in control embryos were used in analysis. A line was drawn manually along the anterior and posterior membranes and in the cytoplasm in three slices within each cell in FIJI version 1.53f51 (ImageJ). Mean intensity along each membrane line was normalized to the cytoplasmic mean intensity, and sfGFP-Vangl2 polarity was presented as the ratio between posterior:anterior mean intensities. Statistical analysis was performed using a two-tailed $t$-test in GraphPad Prism version 9.3.1 (GraphPad Software).

## BB positioning quantification

Basal body position was defined as a ratio between the distance of the basal body from the posterior membrane and the anterior-posterior floor plate cell length normalized to 1. Measurements were done in FIJI version 1.53f51 (ImageJ). Polarity values were divided into three categories: posterior (0.66-1), medial (0.33-0.66) and anterior (0-0.33). Statistical analysis was performed using Kruskal-Wallis test with Dunn's multiple comparisons test in GraphPad Prism version 9.3.1 (GraphPad Software).

## Statistics and reproducibility

Details of statistic testing used in each experiment have been described in the methods section and figure legends. Biological replicates were conducted and they reproduced all findings. Number of replicates is indicated in the figure legends. No sample size calculations were performed. Sample size was limited by the number of embryonic and cell samples available and based partially on our previous experience. No data were excluded from analysis. Embryos were randomly allocated to control and *zGrad* mRNA injected groups by randomly dividing and injecting half of a clutch with mRNA at 1-cell stage. In other experiments samples were not randomized as we used genetic mutant and control animals. In embryonic length quantifications, samples that required genotyping were measured prior to genotyping. In other experiments blinding was not possible as obvious phenotypes or fluorescent marker expression were visible. Source data are provided in the Source Data file.

## Micro-computed tomography (micro-CT)

Fish were fixed in neutral buffered 10% formalin (Sigma) overnight at 4 °C and then mounted in 2% low melt agarose (BioShop) in a plastic vial. Specimens were scanned for 2 h using SkyScan 1275 high-speed Micro-CT scanner (Bruker) with the X-ray power at 45 kVp and 200 μA. All three-dimensional Micro-CT datasets were reconstructed with 14.0 μm isotropic resolution. The images were analyzed using CTVox software (Bruker).

## Scanning electron microscopy (SEM)

7-8 month old fish were euthanized with tricane (500 mg/L), followed by submersion of anesthetized fish in ice water for several minutes. Once morbidity was assured, brains were immediately dissected in cold PBST (1x phosphate saline buffer + 0.25% TritonX). Whole brains were fixed for 2 h in 2% paraformaldehyde and 2% glutaraldehyde in 0.1 M sodium cacodylate buffer (pH7.3), subsequently dissected in half and fixed overnight at 4 °C. Samples were rinsed in 0.1 M sodium cacodylate buffer with 0.2 M sucrose (pH7.3) and brains were gradually dehydrated in an ethanol series. The dissected brains were critical point dried in a Bal-tec CPD030 critical point dryer, mounted on aluminum stubs, gold coated for 15 nm in a Leica ACE200 sputter coater and imaged on a FEI XL30 SEM (Philips).

## Whole-mount in situ hybridization

Antisense RNA probes were prepared using DIG RNA labelling kit (Roche) from linearized DNA. Whole mount RNA in situ hybridizations were performed according to Rose et al.[50]. Samples were genotyped after imaging.

## Reporting summary

Further information on research design is available in the Nature Research Reporting Summary linked to this article.

# Data availability

Raw imaging files are available upon request due to large size. Source data are provided with this paper.

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

## Acknowledgements

We gratefully acknowledge the SickKids' Imaging Facility for assistance with confocal microscopy and image analysis, SickKids' Nanoscale Biomedical Imaging Facility for assistance with SEM sample preparation and imaging, the SickKids' Zebrafish Facility technicians for excellent zebrafish care and Jennica Van Gennip and Chloe Rose for technical assistance. This work was supported, in part, by funding from Canadian Institutes of Health Research (FDN-167285) and the Canada Research Chair program to B.C., and the Sigrid Juselius Foundation to M.J.

## Author contributions

M.J. and C.W.B. performed Vangl2 targeting, M.J. performed all Vangl2 knock-in line validation and Vangl2 imaging experiments and analysis, C.W.B. generated the zGrad transgene, P.G.P. performed the Vangl2 protein analysis and N.W.G. performed the micro-CT and scoliosis analysis. B.C. supervised research studies and M.J. and B.C. wrote the manuscript with input from C.W.B., N.W.G. and P.G.P.

## Competing interests

The authors declare no competing interests.
