## [Peer Review File · Nature Communications]

Live imaging and conditional disruption of native PCP activity using endogenously tagged zebrafish sfGFP-Vangl2REVIEWER COMMENTS

Reviewer #1 (Remarks to the Author):

The PCP pathway has been shown to regulate several essential developmental and homeostatic processes by directing asymmetric modification of the cell cytoskeleton. For example, under the influence of PCP, groups of cells move in a shared direction. Furthermore, cells can position their organelles, such as the basal body, accordingly. However, characterisation of vertebrate PCP at the cellular and molecular level is still lacking, except for snapshots (IF staining) or localisation studies of fluorescently tagged components – which, however, change cell polarity and localisation of other PCP markers due to an overexpression phenotype.

In this manuscript, the authors generate a PCP reporter by CRISPR based knockin of GFP in the *Vangl2* locus to generate a functional GFP-*Vangl2*. This line can serve as a PCP reporter without interfering with the PCP system. The homozygous fish seems similar to the WT fish – and only display a shorter body axis.

In general, this is an exquisite study of PCP in a vertebrate model. However, a significant question in the PCP field seems to be overlooked: Recent publications indicate that paracrine Wnt signalling is dispensable for PCP in invertebrates, i.e. *Drosophila*. However, in vertebrates, the situation is still unclear. It seems that the non-canonical Wnt5 is important to form gradients in some tissues. The authors have generated a precious tool to investigate this paradigm in a vertebrate model organism. I believe it would be essential to investigate PCP signalling in Wnt ligand mutants and after local overexpression of non-canonical Wnt ligands to investigate the influence of paracrine signalling on PCP in a vertebrate tissue. This would increase the value of the publication immediately and could be added easily.

Detailed comments

Figure 1f-i: Can the Rab colocalisation be quantified? A Pearson colocalisation coefficient would be helpful. Ullrich et al. 2005 reported that Wnt11 and e-cadherin colocalises with Rab5c. Do they form a complex with *Vangl2*?

Furthermore, in the image, Rab5c and Rab 11a are labelled as just Rab5 and Rab11; this needs to be changed for clarity.

Line 80: Text refers to 22hpf whilst figure 1b indicates 28hpf.

Line 85: What does the body shortening of the mutants, heterozygotes and trans-heterozygotes indicate in terms of PCP specifically? i.e. why is this the readout used to determine abnormalities?

Line 122: Is it possible that targeting late endosomes for degradation is a result of sfGFP modifications? The author needs to address this and compare it to non-tagged *Vangl2*.

Figure 2: The images of the localisation of *Vangl2* in the floor plate is spectacular, and the authors should be congratulated on these stunning pictures. However, I would suggest improving labelling, e.g., adding anterior and posterior descriptions in images for ease.

Figure 2i: The authors claim that *Vangl2* labels cell protrusions. A recent study has shown that *Vangl2* is essential for the formation of signalling cytonemes formation. Can the authors comment on the nature of these protrusions?

Figure 3: Clearer orientation axes would help the reader. Are there any known posterior markers that could be used to distinguish the anterior and posterior membrane?

Line 211: 'However, in some floorplate cells apically enriched *Vangl2* appeared to extend anteriorly towards neighbouring cells'.

This is hard to make out; could this be pointed out in the figure?

Line 212: 'This bright apical Vangl2 signal was closely associated with BBs labelled with Centrin-mCherry, and with Arl13b-positive cilia'.

Can 'closely associated' be expanded upon? It looks like they are localised next to each other on the membrane rather than co-localised. What could this mean biologically in terms of an association?

Figure 4A: What is the impact of degradation in terms of the protein level?

Figure 4C: Could imaging examples of the basal body position changes described be shown, either in 4C or on 4B.

Figure 4B-D:

How do the mosaic Vangl2 loss and subsequent basal body positioning affect the later development of these embryos?

Also, how does this affect subsequent cell divisions in the floorplate? Can you see any particular cell division phenotypes as a result?

Are there any similarities between any phenotypes seen here (either with basal body abnormalities or later cell division or cell irregularities) to the Vangl2(m209) mutants?

Methods:

Line 276: Why were *trtk50f* mutants chosen rather than the m209 mutants for the cell transplantation studies?

Line 350: Could the cell transplantation methods be expanded upon?

Reviewer #2 (Remarks to the Author):

The manuscript by Jussila et al. describes the role of the core planar cell polarity (PCP) protein Vangl2 in the early zebrafish embryo using GFP knock-in fish along with GFP-targeted conditional knock-down. The authors show that endogenous Vangl2 of GFP-knock-in fish is functional and applicable for time-lapse imaging. The authors also show that GFP-nanobody-mediated zebrafish-optimized protein degradation system (zGrad so-called degron) is feasible in a conditional knockdown in zebrafish.

GFP-knock-in in zebrafish is feasible but very challenging, as opposed to *C. elegans* or *Drosophila*, whereas a conditional degron system is not established based on knocked-in zebrafish. Thus, this paper provides some useful information to the zebrafish community. Despite the importance of visualizing endogenous PCP proteins in time and in space, a scientific advance in the research field of PCP is not significant. The zebrafish-specific degron system has been optimized previously (Yamaguchi, 2019 eLife), although Yamaguchi et al. did not show using endogenously knock-in GFP. Hence, there is only little technical innovation involved.

Overall, reflected by the title and abstract, it is unclear as to what is a novel finding and what is a technical advance to be published in a highly profile journal. Thus, I hesitate to support this manuscript for publication in Nature Communications.

(Specific points)

1. Although the authors suggest that Vangl2 is targeted to late endosomes for degradation (Fig 1f-i), this is not fully supported by the data, due to a lack of quantitative analysis of co-localizations (e.g. Pearson correlation coefficient).

2. Following up, rather, this suggestion leads to the concern that late endosomal localization of Vangl2 could be due to mis-folding of sfGFP tagged endogenous protein, which is to be targeted to

autophagosomes for degradation. Thus, sfGFP knock-in allele could be weakly hypomorphic (Fig 1d). Are Vangl2 puncta co-localized with the autophagosome marker LC3? Or the authors ought to validate the late endosomal localization of Vangl2 using an anti-Vangl2 antibody (Roszko, 2015).

3. There is not significant novelty regarding anterior localization of Vangl2 puncta based on time-lapse movies (supple movies 1-4) using the sfGFP-knock-in fish. What are cellular events/processes upstream of anterior localization of Vangl2 puncta? Also, how does Vangl2 determine the position of the basal body?

Even a clue to answering to these questions is not suggested from time-lapse analysis of endogenous Vangl2. Proper quantitative image analysis is required to suggest that PCP signaling functions in 'specific' subdomains on the anterior membrane.

4. Why does the degron-mediated tissue-specific knockdown lead to mosaic KD of Vangl2 cells in the floor plate? Is it possible to knock down in entire floor plate cells? Or, this degron-mediated tissue-specific knockdown system may not be applicable in zebrafish embryos/larvae in general.

Responses to the reviewers for Jussila et al.

Reviewer #1 (Remarks to the Author):

The PCP pathway has been shown to regulate several essential developmental and homeostatic processes by directing asymmetric modification of the cell cytoskeleton. For example, under the influence of PCP, groups of cells move in a shared direction. Furthermore, cells can position their organelles, such as the basal body, accordingly. However, characterisation of vertebrate PCP at the cellular and molecular level is still lacking, except for snapshots (IF staining) or localisation studies of fluorescently tagged components – which, however, change cell polarity and localisation of other PCP markers due to an overexpression phenotype.

In this manuscript, the authors generate a PCP reporter by CRISPR based knockin of GFP in the *Vangl2* locus to generate a functional GFP-*Vangl2*. This line can serve as a PCP reporter without interfering with the PCP system. The homozygous fish seems similar to the WT fish – and only display a shorter body axis.

In general, this is an exquisite study of PCP in a vertebrate model. However, a significant question in the PCP field seems to be overlooked: Recent publications indicate that paracrine Wnt signalling is dispensable for PCP in invertebrates, i.e. *Drosophila*. However, in vertebrates, the situation is still unclear. It seems that the non-canonical Wnt5 is important to form gradients in some tissues. The authors have generated a precious tool to investigate this paradigm in a vertebrate model organism. I believe it would be essential to investigate PCP signalling in Wnt ligand mutants and after local overexpression of non-canonical Wnt ligands to investigate the influence of paracrine signalling on PCP in a vertebrate tissue. This would increase the value of the publication immediately and could be added easily.

We agree that the nature of the upstream polarizing cue for PCP in vertebrates is an important and exciting question that warrants extensive further investigation, and that our sfGFP-*Vangl2* allele will be instrumental in such studies. Unfortunately, we do not currently house zebrafish *wnt* mutant lines at our facility, and the Canadian Government has imposed strict importation, quarantine and testing regulations that prevent us from accessing and utilizing these fish within a reasonable time frame (Hanwell et al. 2016). However, we have now investigated the consequence of local overexpression of two non-canonical Wnt ligands implicated in zebrafish PCP, namely Wnt5b and Wnt11. The experimental strategy and results are included in a new Supplementary Fig. 5. as well as in the methods section. Briefly, we utilized transient transgenesis in F0 embryos to express a transgene driving a heatshock-inducible Wnt ligand in a mosaic fashion, and transplanted *vangl2^{sfGFP}* cells into these transgene expressing hosts. The transgenes included an IRES-tdTomato fluorescent reporter to identify Wnt5b or Wnt11 expressing cells. We heatshocked embryos at the onset of neural tube morphogenesis, analysed neural tubes for any changes for sfGFP-*Vangl2* localization in cells with Wnt5b or Wnt11 expressing cells in close proximity 5-6 hours later. We quantified the ratio of sfGFP on posterior to anterior membranes, and found no significant change with either ligand. Our results suggest that, at least in this spatial and temporal context, exogenous Wnt5b and Wnt11 cues do not play

an instructive role for PCP. These new experiments, results and discussion are included on lines 220-238 of the revised manuscript.

Hanwell et al. Restrictions on the importation of zebrafish into Canada associated with spring viremia of carp virus. *Zebrafish* **13**, (2016).

Detailed comments

Figure 1f-i: Can the Rab colocalisation be quantified? A Pearson colocalisation coefficient would be helpful. Ullrich et al. 2005 reported that Wnt11 and e-cadherin colocalises with Rab5c. Do they form a complex with Vangl2?

We have added quantification to Supplementary Fig. 2b, paragraphs on lines 115-140 of the revised manuscript. Upon consultation with staff at SickKids' Imaging Facility, we opted against Pearson's correlation coefficient, as it measures colocalization in each pixel. In our case we do not have a 1:1 overlap between the two channels as we often have a Vangl2 positive vesicle surrounded by a Rab signal. Instead, we used Imaris software to segment the sfGFP-Vangl2 signal and respective markers into objects, and quantified the percentage of sfGP-Vangl2 puncta that touch either Rab5c, Rab7, Rab11a or GalT puncta. We also added immunostaining for the autophagy marker LC3 as suggested by Reviewer 2. Quantification demonstrated that there is a comparable amount of Rab5c and Rab7 positive Vangl2 puncta, and we have changed our conclusions in the text accordingly on lines 124-128.

In Ulrich et al. 2005 the authors show that inducing *wnt11* expression in *slb/wnt11* mutants increases the amount cytoplasmic E-cadherin, but not Strabismus (Drosophila homologue of Vangl2), suggesting that Wnt11 does not coregulate E-cadherin and Vangl2. It is possible that Vangl2 and E-cadherin localize to the same Rab5c-positive vesicles, but questions of their colocalization and possible complex formation remain beyond the scope of this study.

Furthermore, in the image, Rab5c and Rab 11a are labelled as just Rab5 and Rab11; this needs to be changed for clarity.

Thank you for noticing this. We have updated the Figure 1. accordingly.

Line 80: Text refers to 22hpf whilst figure 1b indicates 28hpf.

Thank you for noticing this. 28 hpf is correct. We have updated the text accordingly on line 90 of the revised manuscript.

Line 85: What does the body shortening of the mutants, heterozygotes and trans-heterozygotes indicate in terms of PCP specifically? i.e. why is this the readout used to determine abnormalities?

Body shortening is a readout of convergence and extension defects common in PCP mutants. It was also the only visible phenotype we observed in the *vangl2^{sfGFP}* knock-in embryos. For

instance, we detected no neural tube closure defects or basal body mislocalization defects. We have clarified the use of this readout in lines 90-94 of the revised manuscript.

Line 122: Is it possible that targeting late endosomes for degradation is a result of sfGFP modifications? The author needs to address this and compare it to non-tagged Vangl2.

It is possible that the addition of the sfGFP protein to endogenous Vangl2 affects protein folding, stability or function to some degree, resulting in higher-than-normal protein degradation. However, the observation that the levels of cytoplasmic sfGFP-Vangl2 decrease dramatically after the onset of gastrulation would suggest that this is not a property of the fusion protein, but a normal developmental process. To support this, Roszko et al. have reported a similar cytoplasmic Vangl2 signal using an anti-Vangl2 antibody. In addition, as the *vangl2* mutant phenotype is lethal, and fish homozygous for our *vangl2*^{sfGFP} knock-in allele are viable and fertile without any gross phenotypes apart from the shorter body length at embryonic stages, we conclude that the sfGFP-Vangl2 protein must be largely functional. During the revision process a paper describing a mouse tdTomato-Vangl2 knock-in allele was reported (Basta et al. 2021). Mice both hetero and homozygous for this allele show PCP-related phenotypes, with the homozygous embryos displaying variable neural tube closure defects. Therefore, we conclude that our sfGFP-Vangl2 allele is less disruptive in function and likely represents normal Vangl2 localization pattern. We also show in new results that targeted degradation of sfGFP-Vangl2 in multiciliated cells results in adolescent idiopathic scoliosis, which is a phenotype we do not observe in *vangl2*^{sfGFP/sfGFP} fish, further supporting that the sfGFP-Vangl2 protein is not normally degraded in significant amounts.

We agree that detecting endogenous untagged Vangl2 and comparing it with our sfGFP-Vangl2 would be extremely interesting. Unfortunately, there are no commercial zebrafish-specific Vangl2 antibodies currently available. In addition, Reviewer 2. suggested that we compare sfGFP-Vangl2 localization with the autophagy marker LC3 to analyze if Rab7 positive vesicles that we observed could be autophagosomes. New data in Supplementary Fig. 2 a-c shows that sfGFP-Vangl2 does colocalize with LC3 to some extent, but not to the same degree as Rab7. Therefore, we cannot exclude the possibility that the sfGFP tag induces some level of protein misfolding and degradation during gastrulation, but overall, this does not compromise normal development and homeostasis. We have added LC3 analysis to the revised manuscript, and modified our conclusions based on these results on lines 130-140 of the revised manuscript.

Roszko, I., S. Sepich, D., Jessen, J. R., Chandrasekhar, A. & Solnica-Krezel, L. A dynamic intracellular distribution of Vangl2 accompanies cell polarization during zebrafish gastrulation. *Development* **142**, 2508–2520 (2015).
Basta, L. P. *et al.* New mouse models for high resolution and live imaging of planar cell polarity proteins in vivo. *Development* **148**, dev199695 (2021).

Figure 2: The images of the localisation of Vangl2 in the floor plate is spectacular, and the authors should be congratulated on these stunning pictures. However, I would suggest improving labelling, e.g., adding anterior and posterior descriptions in images for ease.

We have added labels to indicate the anterior direction in Fig. 2.

Figure 2i: The authors claim that Vangl2 labels cell protrusions. A recent study has shown that Vangl2 is essential for the formation of signalling cytonemes formation. Can the authors comment on the nature of these protrusions?

We have indeed referred to the paper by Brunt et al. in our discussion on the cellular protrusions we observed in our Supplementary Movies. Brunt et al. does not analyze cytonemes in the zebrafish neuroepithelium, and therefore it is unclear if these structures exist in this tissue type. However, PCP has been linked to the regulation of protrusive activity and the localization of sfGFP-Vangl2 into cellular protrusions supports these earlier observations.

Figure 3: Clearer orientation axes would help the reader. Are there any known posterior markers that could be used to distinguish the anterior and posterior membrane?

We have cropped the images less and added labels for Fig.3 for the anterior direction similar to Fig. 2. Currently the best posterior marker that we are aware of is the posteriorly polarized basal body, which we have used in these experiments.

Line 211: 'However, in some floorplate cells apically enriched Vangl2 appeared to extend anteriorly towards neighbouring cells'.

This is hard to make out; could this be pointed out in the figure?

We have added arrows to make this clearer to Fig. 3a-b. In addition, we added a different colour arrow to Fig. 3c point at the apical enrichment, as well as a dashed line to indicate the position of the unlabelled neighbouring host cell to help the reader orient themselves.

Line 212: 'This bright apical Vangl2 signal was closely associated with BBs labelled with Centrin-mCherry, and with Arl13b-positive cilia'.

Can 'closely associated' be expanded upon? It looks like they are localised next to each other on the membrane rather than co-localised. What could this mean biologically in terms of an association?

As the anterior Vangl2 signal and the posteriorly polarized basal body and cilia are in different cells, we do not suggest that they co-localize. Instead, Vangl2 likely interacts with other PCP coreceptors on the neighboring cell membrane, and the downstream effects of this interaction lead to changes in cytoskeletal organization, apical polarity and/or cell shape that then maintain the basal body posterior positioning. We have clarified this on lines 258-260 of the revised manuscript.

Figure 4A: What is the impact of degradation in terms of the protein level?

We have performed Western blot against GFP and show now in Fig. 4b and Supplementary Fig. 6d that injecting *zGrad* mRNA into *vangl2^{sfGFP/sfGFP}* embryos leads to efficient degradation of the sfGFP-Vangl2 protein. This is discussed on lines 272-273 of the revised manuscript.

Figure 4C: Could imaging examples of the basal body position changes described be shown, either in 4C or on 4B.

We have indicated mispolarized basal bodies in Fig. 4c (previously Fig. 4b). We did not want to give specific examples of basal bodies in cells in the different conditions analysed in Fig. 4d-e (previously Fig. 4c-d) as there is a range of phenotypes within each. This could bias the reader to incorrectly assume that we observed only one type of polarized basal body positioning in each condition.

Figure 4B-D:

How do the mosaic Vangl2 loss and subsequent basal body positioning affect the later development of these embryos?

Excellent question, and we have included new data into Figure. 5 describing exciting embryonic and adult phenotypes of the *vangl2^{sfGFP};Tg(β actin2::loxP-mCherry-STOP-loxP-zGrad);Tg(foxl1a::iCre)* fish. Degradation of sfGFP-Vangl2 in *foxl1a*-expressing motile ciliated cell lineages results in embryonic body curvatures and causes adolescent idiopathic-like scoliosis post-embryonically. This is an exciting finding as it builds on our previous observation that *ptk7* is required specifically in FoxJ1a-positive cells to drive idiopathic scoliosis (IS) pathogenesis in zebrafish (Hayes et al. 2014, Grimes et al. 2016, Van Gennip et al. 2018). Ptk7 is a modulator of both a modulator of both non-canonical/PCP and Wnt/beta-catenin signalling, and therefore it was not clear which of the two pathways was perturbed in the mutants. As Vangl2 is a core and specific component of PCP signalling, the phenotype of the *zGrad;foxl1a::iCre;vangl2^{sfGFP}* fish confirms PCP as a driver of IS. Our findings firmly establish a key role for PCP signalling in spine morphogenesis, and provides new pathogenic mechanisms that may be relevant to human scoliosis. Indeed, variants in the core PCP genes *PTK7* and *VANGL1* have been identified in human IS patients. These exciting new findings have been included on lines 292-312 of the revised manuscript.

Hayes, M. *et al.* Ptk7 mutant zebrafish models of congenital and idiopathic scoliosis implicate dysregulated Wnt signalling in disease. *Nat. Commun.* **5**, 1–11 (2014).

Grimes, D. T. *et al.* Zebrafish models of idiopathic scoliosis link cerebrospinal fluid flow defects to spine curvature. *Science*. **352**, 1341–1344 (2016).

Van Gennip, J. L. M., Boswell, C. W. & Ciruna, B. Neuroinflammatory signals drive spinal curve formation in zebrafish models of idiopathic scoliosis. *Sci. Adv.* **4**, 1–12 (2018).

Also, how does this affect subsequent cell divisions in the floorplate? Can you see any particular cell division phenotypes as a result?

Basal bodies are modified centrioles that function specifically in nucleating the cilia. As PCP has been specifically linked to basal body positioning and cilia polarization in the floor plate, we did not analyse floor plate cell divisions.

Are there any similarities between any phenotypes seen here (either with basal body abnormalities or later cell division or cell irregularities) to the Vangl2(m209) mutants?

Both *vangl2* mutant lines used in the study, *trik50f* and *tri^{m209}*, have been shown by Borovina et al. and Donati et al. to display basal body polarization defects comparable to our *vangl2^{sfGFP};Tg(βactin2::loxP-mCherry-STOP-loxP-zGrad);Tg(foxa::iCre)* fish.

Borovina, A., Superina, S., Voskas, D. & Ciruna, B. Vangl2 directs the posterior tilting and asymmetric localization of motile primary cilia. *Nat. Cell Biol.* **12**, (2010).

Donati, A., Anselme, I., Schneider-Maunoury, S. & Vesque, C. Planar polarization of cilia in the zebrafish floorplate involves Par3-mediated posterior localization of highly motile basal bodies. *Development* **148**, (2021).

Methods:

Line 276: Why were *trik50f* mutants chosen rather than the m209 mutants for the cell transplantation studies?

This was done simply for practical reasons. We had completed cell transplantation experiments earlier using the *trik50f* line, selecting for host embryos demonstrating obvious Vangl2 loss-of-function phenotypes. For the analysis of sfGFP-Vangl2 fusion protein functionality, we required a robust molecular genotyping protocol so that we could identify sfGFP-Vangl2/Vangl2 mutant trans-heterozygous embryos (which are phenotypically normal). Therefore, we chose to use the *tri^{m209}* line because robust genotyping primers/protocols exist for the *m209* mutation.

Line 350: Could the cell transplantation methods be expanded upon?

We have expanded on the cell transplantation method in the Methods section on lines 412-420 of the revised manuscript and added a reference to a very thorough publication by Kemp et al. in *Journal of Visualized Experiments* describing the process.

Kemp, H. A., Carmany-Rampey, A. & Moens, C. Generating Chimeric Zebrafish Embryos by Transplantation. *JoVE* **e1394**, (2009).

Reviewer #2 (Remarks to the Author):

The manuscript by Jussila et al. describes the role of the core planar cell polarity (PCP) protein Vangl2 in the early zebrafish embryo using GFP knock-in fish along with GFP-targeted conditional knock-down. The authors show that endogenous Vangl2 of GFP-knock-in fish is functional and applicable for time-lapse imaging. The authors also show that GFP-nanobody-mediated zebrafish-optimized protein degradation system (zGrad so-called degnon) is feasible in a conditional knockdown in zebrafish.

GFP-knock-in in zebrafish is feasible but very challenging, as opposed to *C. elegans* or *Drosophila*, whereas a conditional degnon system is not established based on knocked-in zebrafish. Thus, this paper provides some useful information to the zebrafish community. Despite the importance of visualizing endogenous PCP proteins in time and in space, a scientific advance in the research field of PCP is not significant. The zebrafish-specific degnon system has been optimized previously (Yamaguchi, 2019 eLife), although Yamaguchi et al. did not show using endogenously knock-in GFP. Hence, there is only little technical innovation involved.

We thank the reviewer for their critical review of our work. Nevertheless, we believe interest in our findings will extend far beyond the zebrafish community, as we present the only functional endogenous fluorescently-tagged PCP reporter in a vertebrate system. Notably, a recent publication by Basta et al. describes a tdTomato-Vangl2 knock-in allele in mouse. However, this fusion protein is not fully functional, as both heterozygous and homozygous mice carrying this allele show variable phenotypes. Our demonstration of a functional sfGFP-Vangl2 zebrafish allele highlights the importance of using a fluorophore that does not disrupt protein function when targeting endogenous proteins, and thus serves as a point of reference for the wider scientific community interested in generating fluorescent knock-in alleles in different model systems.

Basta et al. New mouse models for high resolution and live imaging of planar cell polarity proteins *in vivo*. *Development* **148**, (2021).

Moreover, our endogenous Vangl2 knock-in allele validates previous observations on anterior-posterior planar polarity of the neuroepithelium, detected in multiple model systems using exogenous reporters that have limited credibility due to potential overexpression artefacts. Additionally, we describe Vangl2 polarity at the level of single cells at unprecedented resolution. Our results also show that PCP is required non-cell autonomously not just to establish but also to maintain basal body polarity, something that has been difficult to demonstrate without conditional tools. Furthermore, as described in our response to Reviewer#1 and below, our revised manuscript includes exciting new data demonstrating a late-onset scoliosis phenotype in *vangl2^{sfGFP};Tg(β actin2::loxP-mCherry-STOP-loxP-zGrad);Tg(foxl1a::iCre)* fish. This highlights the utility of fluorescent knock-in alleles, not only for visualizing protein localization, but as an alternative method for the analysis of conditional gene function in zebrafish. This is a critical technical advance in the field of zebrafish research, where the lack of conditional mutants has hindered the research of postembryonic gene function for embryonic lethal mutations. In the

future, our *vangl2^{sfGFP}* allele will allow the continuation of analysis of endogenous PCP activity during different developmental processes as well as, for the first time, allowing tissue and cell-type specific interrogation of PCP function in adult homeostasis and disease.

Overall, reflected by the title and abstract, it is unclear as to what is a novel finding and what is a technical advance to be published in a highly profile journal. Thus, I hesitate to support this manuscript for publication in Nature Communications.

In our revised manuscript, we are excited to include new data into Figure. 5 and to lines 292-312 describing novel embryonic and adult phenotypes for the *vangl2^{sfGFP};Tg(β actin2::loxP-mCherry-STOP-loxP-zGrad);Tg(foxj1a::iCre)* fish. Degradation of sfGFP-Vangl2 in *foxJ1a*-expressing motile ciliated cell lineages results in embryonic body curvatures and causes adolescent idiopathic-like scoliosis post-embryonically. This is an exciting finding as it builds on our previous observation that *ptk7* is required specifically in FoxJ1a-positive cells to drive idiopathic scoliosis (IS) pathogenesis in zebrafish (Hayes et al. 2014, Grimes et al. 2016, Van Gennip et al. 2018). Ptk7 is a modulator of both non-canonical/PCP and Wnt/beta-catenin signalling, and therefore it was not clear which of the two pathways was perturbed in the mutants. As Vangl2 is a core and specific component of PCP signalling, the phenotype of the *vangl2^{sfGFP};Tg(β actin2::loxP-mCherry-STOP-loxP-zGrad);Tg(foxj1a::iCre)* fish confirms PCP as a driver of IS. Our findings firmly establish a role for PCP in spine morphogenesis, and provides new pathogenic mechanisms that may be relevant to human scoliosis. Notably, variants in both PTK7 and VANGL1 have been identified in human IS patients, extending interest in our findings beyond the zebrafish scoliosis community.

Hayes, M. *et al.* Ptk7 mutant zebrafish models of congenital and idiopathic scoliosis implicate dysregulated Wnt signalling in disease. *Nat. Commun.* **5**, 1–11 (2014).

Grimes, D. T. *et al.* Zebrafish models of idiopathic scoliosis link cerebrospinal fluid flow defects to spine curvature. *Science*. **352**, 1341–1344 (2016).

Van Gennip, J. L. M., Boswell, C. W. & Ciruna, B. Neuroinflammatory signals drive spinal curve formation in zebrafish models of idiopathic scoliosis. *Sci. Adv.* **4**, 1–12 (2018).

(Specific points)

1. Although the authors suggest that Vangl2 is targeted to late endosomes for degradation (Fig 1f-i), this is not fully supported by the data, due to a lack of quantitative analysis of colocalizations (e.g. Pearson correlation coefficient).

As addressed above in response to Reviewer #1, we have now added quantitative analysis of colocalization to Supplementary Fig. 2b. Upon consultation with staff at SickKids' Imaging Facility, we opted against Pearson's correlation coefficient, as it measures colocalization in each pixel. In our case we do not have a 1:1 overlap between the two channels as we often have a Vangl2 positive vesicle surrounded by a Rab signal. Instead, we used Imaris software to segment the sfGFP-Vangl2 signal and respective markers into objects, and quantified the percentage of sfGP-Vangl2 puncta that touch either Rab5c, Rab7, Rab11a or GalT puncta. We also added immunostaining for the autophagy marker LC3 as suggested by Reviewer 2. Quantification demonstrated that there is a comparable amount of Rab5c and Rab7 positive Vangl2 puncta, and

we have changed our conclusions in the text accordingly on lines 124-128 of the revised manuscript.

2. Following up, rather, this suggestion leads to the concern that late endosomal localization of Vangl2 could be due to mis-folding of sfGFP tagged endogenous protein, which is to be targeted to autophagosomes for degradation. Thus, sfGFP knock-in allele could be weakly hypomorphic (Fig 1d). Are Vangl2 puncta co-localized with the autophagosome marker LC3? Or the authors ought to validate the late endosomal localization of Vangl2 using an anti-Vangl2 antibody (Roszko, 2015).

It is possible that the addition of the sfGFP protein to endogenous Vangl2 affects protein folding, stability or function to some degree, resulting in higher-than-normal protein degradation. However, the observation that the levels of cytoplasmic sfGFP-Vangl2 decrease dramatically after the onset of gastrulation would suggest that this is not a property of the fusion protein, but a normal developmental process. To support this, Roszko et al. report a similar cytoplasmic Vangl2 signal using an anti-Vangl2 antibody. In addition, as the *vangl2* mutant phenotype is lethal, and fish homozygous for our *vangl2^{sfGFP}* knock-in allele are viable and fertile without any gross phenotypes apart from the shorter body length at embryonic stages, we conclude that the sfGFP-Vangl2 protein must be largely functional. During the revision process a paper describing a mouse tdTomato-Vangl2 knock-in allele was reported (Basta et al. 2021). Mice both hetero and homozygous for this allele show PCP-related phenotypes, with the homozygous embryos displaying variable neural tube closure defects. Therefore, we conclude that our sfGFP-Vangl2 allele is less disruptive in function and likely represents normal Vangl2 localization pattern. We also show with new results that targeted degradation of sfGFP-Vangl2 in multiciliated cells results in adolescent idiopathic scoliosis, which is a phenotype we do not observe in *vangl2^{sfGFP/sfGFP}* fish, further supporting that the sfGFP-Vangl2 protein is not normally degraded in significant amounts.

As per reviewer suggestions, we used an LC3 antibody to analyse its colocalization with Vangl2. New data in Supplementary Fig. 2 a-c shows that sfGFP-Vangl2 does colocalize with LC3 to some extent, but not to the same degree as with the late endosome marker Rab7. Therefore, although we cannot exclude the possibility that the sfGFP tag induces some level of protein misfolding and degradation during gastrulation, overall this does not compromise normal development and homeostasis. We agree that detecting endogenous Vangl2 and comparing it with our sfGFP-Vangl2 would be extremely interesting. Unfortunately, the antibody used in Roszko et al. is no longer commercially available, and we are not aware of any commercial antibody specific for zebrafish Vangl2. We have modified our conclusions based on these results on lines 130-140 of the revised manuscript.

Basta, L. P. *et al.* New mouse models for high resolution and live imaging of planar cell polarity proteins in vivo. *Development* **148**, dev199695 (2021).

3. There is not significant novelty regarding anterior localization of Vangl2 puncta based on time-lapse movies (supple movies 1-4) using the sfGFP-knock-in fish. What are cellular events/processes upstream of anterior localization of Vangl2 puncta? Also, how does Vangl2 determine the position of the basal body? Even a clue to answering to these questions is not suggested from time-lapse analysis of endogenous Vangl2. Proper quantitative image analysis is required to suggest that PCP signaling functions in 'specific' subdomains on the anterior membrane.

Upon suggestion from Reviewer 1, we have now investigated the instructive roles for non-canonical Wnt ligands, Wnt5b and Wnt11, on the anterior localization of Vangl2. We have included this new data to Supplementary Fig. 5 and on lines 220-238 of the revised manuscript. Although the downstream mechanisms by which Vangl2 control basal body positioning are beyond the scope of this study, a recent publication by Donati et al. demonstrated interesting interactions between PCP and Vangl2 in floor plate cells, the posterior enrichment of Par3 and cytoskeletal mechanical forces that are required for posterior docking of basal bodies. As Vangl2 likely interacts with other PCP coreceptors on opposing cell membranes, we believe this local PCP activity could regulate similar downstream mechanisms in a non-cell autonomous manner, and we have expanded on our discussion of this matter on lines 258-260 of the revised manuscript. We agree that our time-lapse movies are only descriptive in nature. However, heterogeneity of individual imaged cells and the low-throughput nature of zebrafish cell transplantation experiments blocked our ability to yield meaningful statistical data from quantitative analyses. We respectively submit that our time-lapse movies will nevertheless be interesting and useful to readers and we have toned down the conclusion regarding specific subdomains on lines 181-182 of the revised manuscript.

Donati, A., Anselme, I., Schneider-Maunoury, S. & Vesque, C. Planar polarization of cilia in the zebrafish floor-plate involves Par3-mediated posterior localization of highly motile basal bodies. *Development* **148**, (2021).

4. Why does the degron-mediated tissue-specific knockdown lead to mosaic KD of Vangl2 cells in the floor plate? Is it possible to knock down in entire floor plate cells? Or, this degron-mediated tissue-specific knockdown system may not be applicable in zebrafish embryos/larvae in general.

We believe the mosaicism is a result of the recombination and degradation kinetics. In lineage tracing experiments using the cre recombinase, efficient activity is sometimes observed only 24-36 hours after induction of cre expression (Carney and Mosimann 2018). In our system, after the successful stop cassette excision by the FoxJ1a-driven cre recombinase, zGrad needs to be transcribed and translated to sufficient amounts to degrade sfGFP-Vangl2. We did observe that at 24 hours there were still significant amounts of floor plate sfGFP-Vangl2 in the experimental embryos. At 48 hours when we performed our analysis, some embryos had floor plates completely devoid of sfGFP-Vangl2, and even the cells with sfGFP-Vangl2 left had weaker fluorescent signal than control floor plate cells. We agree that this system is not ideal for studying rapid events in zebrafish early development. However, it presents a tool to study gene function past embryonic development in juvenile stages and in adults, which in zebrafish has

been a true bottleneck in bringing it to the next level as a model organism for adult tissue homeostasis, disease and regeneration. This is highlighted by new results in Fig. 5 describing the adult phenotype of the *vangl2^{sfGFP};Tg(β actin2::loxP-mCherry-STOP-loxP-zGrad);Tg(foxj1a::iCre)* fish. As noted in our response above, our new results confirm loss of PCP in motile-ciliated cell lineages as a driver for idiopathic-like scoliosis, implicating new pathogenic mechanisms that may be relevant to human scoliosis. This is an exciting technical advancement that expands the toolbox of the zebrafish model system.

Carney, T.J. & Mosimann, C. Switch and trace: recombinase genetics in zebrafish. *Trends in Genetics* **34**, (2018).

REVIEWER COMMENTS

Reviewer #1 (Remarks to the Author):

In the revised version, the authors have addressed most of my questions. Furthermore, the authors have added new results that targeted degradation of Vangl2 can lead to an idiopathic scoliosis phenotype. This is very interesting.

Specific points:

Fig. 1: I agree with reviewer2; it is still unclear to me how the sfGFP-Vangl2 can be compared to the localisation of the endogenous protein. Is sfGFP-Vangl2 routed intracellularly to similar compartments like the endogenous Vangl2? This would be important to show to strengthen the physiological function of sfGFP-Vangl2. Here, I refer to Fig. 1e, which shows significant intracellular accumulations of sfGFP-Vangl2. No antibody staining against the endogenous Vangl2 has been provided. The analysis with LC3 is also interesting; however, not entirely convincing. Therefore, the question of whether tagging leads to misrouting to late endosomes/autophagosomes could not be fully addressed. A detailed discussion needs to be provided.

Fig. 4a: Here, the body length of the sfGFP-Vangl2 + zGrad injected embryos needs to be compared to the Vangl2 m209 mutants, as displayed in Fig. 1d. Is there the possibility of investigating the length of a maternal-zygotic m209 mutant?

Fig.5: Here, the finding that sfGFP-Vangl2 + zGrad shows an idiopathic scoliosis phenotype is exciting and adds substantial value to the manuscript. However, this observation needs further exploration. The authors need to quantify their observations. How often and in which genetic background is such a phenotype observed? Does a heterozygous fish lead to a weaker phenotype?

Furthermore, a mechanistic explanation for the phenotype should be provided. The authors show that Vangl2 is required in the motile-ciliated floor plate cells (Fig. 3). Earlier work of the author and others has suggested that a lack or malfunction of motile cilia function can be linked to idiopathic scoliosis phenotype. For example, the authors could provide data showing the localisation of a cilia marker such as Arl13b in the sfGFP-Vangl2 + zGrad. Such a data set would strengthen the link between the tissue-specific depletion of Vangl2 function and the formation/function of motile cilia in the foxj1a positive cells and the idiopathic scoliosis phenotype.

Furthermore, I disagree with the statement: "Our results demonstrate a key role of PCP signalling in spine morphogenesis" (line 309). The experiment does only provide evidence that the Vangl2 function is required. Further experimental evidence is needed if the authors want to stick to this statement, for example, showing that other Wnt/PCP components display a similar function.

Reviewer #2 (Remarks to the Author):

The revised manuscript by Jussila et al. has strengthened the two points. 1) The non-canonical Wnt5b and Wnt11 ligands are unlikely to be upstream signals that confer anterior localization of Vangl2 in the zebrafish neuroepithelium. 2) Motile-ciliated cell lineage-specific knockout of vangle2 causes the spinal curvature phenotype at later stages, which is reminiscent of idiopathic scoliosis in humans.

Together with the majority of the reviewers' concerns being clarified to a satisfactory degree, I would be pleased to support this manuscript for publication in Nature Communications in the present form.

Point-by-Point Response to Reviewer Comments

Jussila et al. Live imaging and conditional disruption of native PCP activity using endogenously tagged zebrafish sfGFP-Vangl2

Our response to comments, and list of revisions made to address reviewer concerns are detailed below.

Reviewer #1 (Remarks to the Author):

In the revised version, the authors have addressed most of my questions. Furthermore, the authors have added new results that targeted degradation of Vangl2 can lead to an idiopathic scoliosis phenotype. This is very interesting.

We thank Reviewer #1 for their supportive comments.

Specific points:

Fig. 1: I agree with reviewer2; it is still unclear to me how the sfGFP-Vangl2 can be compared to the localisation of the endogenous protein. Is sfGFP-Vangl2 routed intracellularly to similar compartments like the endogenous Vangl2? This would be important to show to strengthen the physiological function of sfGFP-Vangl2. Here, I refer to Fig. 1e, which shows significant intracellular accumulations of sfGFP-Vangl2. No antibody staining against the endogenous Vangl2 has been provided. The analysis with LC3 is also interesting; however, not entirely convincing. Therefore, the question of whether tagging leads to misrouting to late endosomes/autophagosomes could not be fully addressed. A detailed discussion needs to be provided.

Regarding this comment on sfGFP-Vangl2 localization, Reviewer 2 originally raised the concern and was satisfied with additional data provided. Unfortunately, the current lack of zebrafish Vangl2 specific antibodies makes further experimentation impossible. We have, however, reworked discussion on this section to emphasize 1) Because cytoplasmic sfGFP-Vangl2 puncta appear only transiently in development, disappearing at later embryonic stages (Fig. 2a, Fig. 3 a-b), this suggests that they do not represent an innate misfolded protein state, but may rather reflect a normal developmental process, and 2) That in accordance, immunohistochemical analysis of endogenous Vangl2 localization has also identified cytoplasmic punctate staining at gastrula stages (Roszko et al. 2015). This text has been added on page 4, second paragraph of the revised manuscript.

Roszko, I., S. Sepich, D., Jessen, J. R., Chandrasekhar, A. & Solnica-Krezel, L. A dynamic intracellular distribution of Vangl2 accompanies cell polarization during zebrafish gastrulation. *Development* **142**, 2508–2520 (2015).

Fig. 4a: Here, the body length of the sfGFP-Vangl2 + zGrad injected embryos needs to be compared to the Vangl2 m209 mutants, as displayed in Fig. 1d. Is there the possibility of investigating the length of a maternal-zygotic m209 mutant?

As instructed by the editor, “we would not require the addition of m/z Vangl2 m209 mutant analysis as requested for Figure 4.” We have therefore not responded to this comment.

Fig.5: Here, the finding that sfGFP-Vangl2 + zGrad shows an idiopathic scoliosis phenotype is exciting and adds substantial value to the manuscript. However, this observation needs further exploration. The authors need to quantify their observations. How often and in which genetic background is such a phenotype observed? Does a heterozygous fish lead to a weaker phenotype?

We have now added this information on page 8, second paragraph of the revised manuscript. To summarize:

- 1) 100% of *vangl2^{sfGFP/sfGFP};Tg(βactin2::loxP-mCherry-STOP-loxP-zGrad);Tg(foxj1a::iCre)* embryos that exhibit axial curvatures and survive to adulthood develop obvious spinal curvature (n=84).
- 2) Spinal curvatures appeared independent of embryonic defects, as conditional mutant embryos with no obvious embryonic phenotype also developed scoliosis at juvenile stages (n = 7/9).
- 3) *vangl2^{sfGFP/sfGFP};Tg(βactin2::loxP-mCherry-STOP-loxP-zGrad)* control animals (which lacked Cre recombinase expression) do not develop scoliosis (n=134)
- 4) *vangl2^{sfGFP/+};Tg(βactin2::loxP-mCherry-STOP-loxP-zGrad);Tg(foxj1a::iCre)* control animals (which experience only heterozygous loss of sfGFP-Vangl2) do not develop scoliosis (n = 85)
- 5) Additionally, we do not detect scoliosis in fish heterozygous for the *vangl2^{m209}* loss-of-function allele, nor in *vangl2^{sfGFP/m209}* trans-heterozygote fish.

These observations demonstrate that the scoliosis phenotype results from degradation of sfGFP-Vangl2 protein in the absence of wild-type Vangl2, specifically within FoxJ1a-positive motile ciliated cell lineages.

Furthermore, a mechanistic explanation for the phenotype should be provided. The authors show that Vangl2 is required in the motile-ciliated floor plate cells (Fig. 3). Earlier work of the author and others has suggested that a lack or malfunction of motile cilia function can be linked to idiopathic scoliosis phenotype. For example, the authors could provide data showing the localisation of a cilia marker such as Arl13b in the sfGFP-Vangl2 + zGrad. Such a data set would strengthen the link between the tissue-specific depletion of Vangl2 function and the formation/function of motile cilia in the foxj1a positive cells and the idiopathic scoliosis phenotype.

We thank Reviewer #1 for this suggestion, and we have now included significant new data investigating a mechanistic explanation for the scoliosis phenotype in conditional sfGFP-Vangl2 mutants. Unfortunately, we were unable to use cilia markers such as *Arl13b* at stages relevant to scoliosis as mRNA injections do not persist past embryonic development, and our *Arl13b* transgenic line (Borovina et al. 2010) has a GFP fluorophore so it cannot be combined with the sfGFP-Vangl2 line. Instead, we've looked at two phenotypes previously linked to scoliosis in zebrafish. These include 1) loss of ependymal cell cilia and 2) ectopic Scospondin accumulation and disruption of Reissner fiber formation within ventricles of the brain. Both of these show similar changes observed previously in scoliotic *ptk7a* mutant zebrafish as well as multiple additional zebrafish IS models, pointing at shared mechanisms in motile cilia and cerebrospinal fluid homeostasis underlying idiopathic scoliosis. An extensive description of this new data can be found on page 8 and 9 of the revised manuscript (new section entitled "Idiopathic scoliosis is associated with ependymal cell cilia and Reissner fiber defects"), as well as in new Figures 6 and 7.

Borovina, A., Superina, S., Voskas, D., and Ciruna, B. Vangl2 directs the posterior tilting and asymmetric localization of motile primary cilia. *Nat. Cell Biol.* **12**, 407–412 (2010).

Furthermore, I disagree with the statement: "Our results demonstrate a key role of PCP signalling in spine morphogenesis" (line 309). The experiment does only provide evidence that the Vangl2 function is required. Further experimental evidence is needed if the authors want to stick to this statement, for example, showing that other Wnt/PCP components display a similar function.

We agree that analysis of the relative contributions of all PCP pathway members to scoliosis is an interesting question. Unfortunately, most zebrafish PCP mutants are embryonic lethal, preventing functional analysis in postembryonic phenotypes. Furthermore, we do not have similar GFP-tagged lines available for PCP components other than Vangl2 to use with zGrad for conditional tissue-specific degradation and generating them is beyond the scope of this manuscript.

However, in the revised manuscript we have now highlighted parallels between our conditional sfGFP-Vangl2 mutant and published *ptk7a* mutant phenotypes – both of which disrupt PCP signaling and produce idiopathic-like scoliosis phenotypes. Although *Ptk7a* regulates both non-canonical Wnt/PCP and canonical Wnt/ β -catenin signalling pathways, the similarities between *ptk7a* and *vangl2* scoliosis phenotypes are remarkable: conditional loss of *ptk7a* specifically with *foxj1a* positive lineages within the brain is sufficient to cause scoliosis, and Idiopathic scoliosis in *ptk7a* mutants is also associated with ependymal cell cilia and Reissner fiber defects. Taken together, these results suggest that scoliosis in conditional *vangl2*^{sfGFP/sfGFP} mutants may be mechanistically linked IS in *ptk7a* mutant models, implicating a key role for PCP signalling in spine morphogenesis and supporting growing evidence that mutations in core PCP genes *PTK7* and *VANGL1*, identified in human patients with adolescent idiopathic scoliosis, may be

functionally linked with spinal curvature. These changes have been made to the final section of the results section.

To further address Reviewer #1's concern, however, we have changed our concluding statement to that of:

our findings "establish a critical role for Vangl2 within motile-ciliated cell lineages for normal zebrafish spine development, possibly linking PCP defects with the pathogenesis of idiopathic scoliosis".

Reviewer #2 (Remarks to the Author):

The revised manuscript by Jussila et al. has strengthened the two points. 1) The non-canonical Wnt5b and Wnt11 ligands are unlikely to be upstream signals that confer anterior localization of Vangl2 in the zebrafish neuroepithelium. 2) Motile-ciliated cell lineage-specific knockout of vangle2 causes the spinal curvature phenotype at later stages, which is reminiscent of idiopathic scoliosis in humans.

Together with the majority of the reviewers' concerns being clarified to a satisfactory degree, I would be pleased to support this manuscript for publication in Nature Communications in the present form.

We thank Reviewer #2 for their kind support of the publication of this manuscript.

REVIEWERS' COMMENTS

Reviewer #1 (Remarks to the Author):

The manuscript has been greatly improved. The additional data are excellent. I, therefore, endorse this manuscript for publication.

Point-by-Point Response to Reviewer Comments

Jussila et al. Live imaging and conditional disruption of native PCP activity using endogenously tagged zebrafish sfGFP-Vangl2

Our response to reviewer's comments are detailed below.

REVIEWERS' COMMENTS

Reviewer #1 (Remarks to the Author):

The manuscript has been greatly improved. The additional data are excellent. I, therefore, endorse this manuscript for publication.

We thank Reviewer #1 for their supportive comments. No further revisions were required.